# Mapping the adaptive landscape of Batesian mimicry using 3D-printed stimuli

Christopher H. Taylor[1✉], David James George Watson[1], John Skelhorn[2], Danny Bell[1], Simon Burdett[1], Aoife Codyre[1], Kathryn Cooley[1], James R. Davies[1,3], Joshua Joseph Dawson[1], Tahiré D'Cruz[1], Samir Raj Gandhi[1,4], Hannah J. Jackson[1], Rebecca Lowe[1], Elizabeth Ogilvie[1], Alexandra Lei Pond[1], Hallie Rees[1], Joseph Richardson[1], Joshua Sains[1], Francis Short[1], Christopher Brignell[5], Gabrielle L. Davidson[6,7], Hannah M. Rowland[8,9], Mark East[10], Ruth Goodridge[10], Francis Gilbert[1] & Tom Reader[1]

In a classic example of adaptation, harmless Batesian mimics gain protection from predators through resemblance to one or more unpalatable models[1,2]. Mimics vary greatly in accuracy, and explaining the persistence of inaccurate mimics is an ongoing challenge for evolutionary biologists[3,4]. Empirical testing of existing hypotheses is constrained by the difficulty of assessing the fitness of phenotypes absent among extant species, leaving large parts of the adaptive landscape unexplored[5]—a problem affecting the study of the evolution of most complex traits. Here, to address this, we created mimetic phenotypes that occupy hypothetical areas of trait space by morphing between 3D images of real insects (flies and wasps), and tested the responses of real predators to high-resolution, full-colour 3D-printed reproductions of these phenotypes. We found that birds have an excellent ability to learn to discriminate among insects on the basis of subtle differences in appearance, but this ability is weaker for pattern and shape than for colour and size traits. We found that mimics gained no special protection from intermediate resemblance to multiple model phenotypes. However, discrimination ability was lower in some invertebrate predators (especially crab spiders and mantises), highlighting that the predator community is key to explaining the apparent inaccuracy of many mimics.

Batesian mimics gain protection when predators treat them as defended 'models' despite being palatable prey[1,2]. As this deception of predators relies on a degree of perceived similarity, increasing resemblance should give a higher probability of misidentification. Yet mimics vary greatly in accuracy[3,6], raising the question of what stops ever-greater mimetic accuracy from evolving[7]. Numerous theoretical explanations[8] have proposed functional trade-offs affecting mimetic appearance[9,10] and factors that might cause relaxed selection for accuracy[4] such as predators' inability to detect differences between mimics and models[11] or reduced motivation to discriminate[12]. Many of these hypotheses are untested experimentally in realistic systems, and there is no consensus about the causes of variation in mimetic accuracy.

The expected outcomes of selection on visual adaptations depend on the specific characteristics of signallers and receivers in each study system[13]. Different visual receivers interpret the same colour patterns in different ways[14]; some features are more easily associated with a reward than others by a given receiver[15]; and systems with more stimulus types elicit more generalized responses[16,17]. Thus, while experiments using abstract stimuli can illuminate general principles, we must test these principles in realistic systems. However, when we

observe extant species, we see only small sections of the adaptive landscape, and miss an opportunity to examine the fitness of phenotypes that do not currently exist. One successful solution to this issue is to manipulate existing phenotypes, for example, by painting or covering real organisms to change their appearance[18], or creating artificial replicas[19], but the manipulations involved tend to be limited in range and realism[19–21].

To overcome these limitations, we generated stimuli combining the relevance and realism of working with real insects, full three-dimensional (3D) representation and the power to manipulate fine details of visual phenotypes. Hoverflies (Syrphidae) are a classic study system to provide reference points for our stimuli, with Batesian mimicry of wasps (Vespidae) varying across species from near-perfect, through approximate, to non-existent mimicry[7] (Extended Data Table 1). We used 3D scans of real model (wasp) and mimic (hoverfly) species as starting points to define axes of variation within multivariate phenotypic space, and generated gradients of mimetic similarity by smoothly manipulating visual traits (shape, colour, pattern and size) along those axes. We then used additive manufacturing (3D printing) to turn these images into physical stimuli, enabling us to explore the

[1]School of Life Sciences, University of Nottingham, Nottingham, UK. [2]Biosciences Institute, Faculty of Medical Sciences, Newcastle University, Newcastle upon Tyne, UK. [3]School of Biological Sciences, University of Bristol, Bristol, UK. [4]The Jolly Geographer, Babraham, UK. [5]School of Mathematical Sciences, University of Nottingham, Nottingham, UK. [6]Department of Psychology, University of Cambridge, Cambridge, UK. [7]School of Biological Sciences, University of East Anglia, Norwich, UK. [8]Predators and Toxic Prey Research Group, Max Planck Institute for Chemical Ecology, Jena, Germany. [9]Department of Evolution, Ecology and Behaviour, Institute of Infection, Veterinary and Ecological Sciences, University of Liverpool, Liverpool, UK. [10]Faculty of Engineering, University of Nottingham, Nottingham, UK. ✉e-mail: c.taylor@nottingham.ac.uk

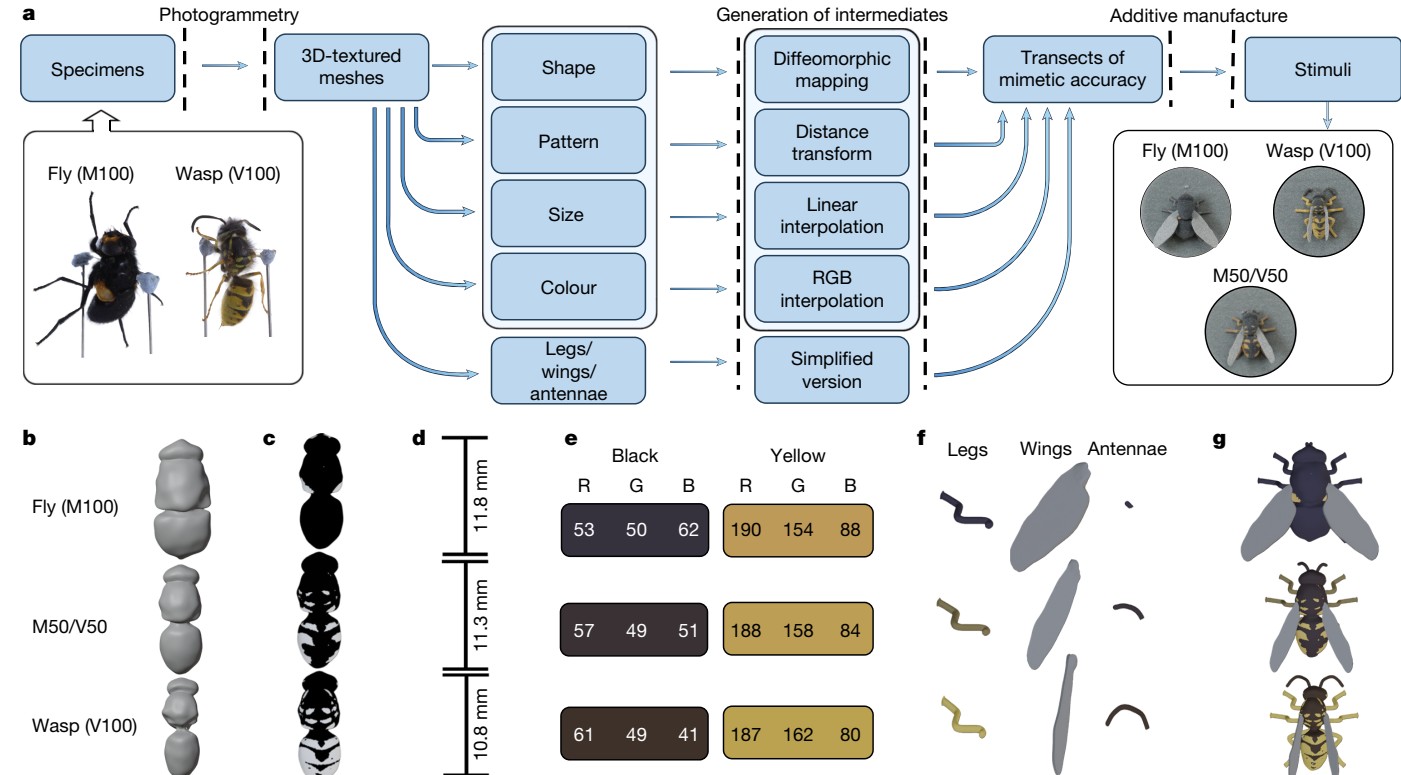

**Fig. 1 | Overview of the methods used to generate artificial mimetic stimuli.** Illustrations are based on an axis of similarity from the fly *M. meridiana* (M100) to the wasp *V. vulgaris* (V100) through a 50% intermediate (M50/V50). **a**, Flowchart of the methodology. Real insect specimens were scanned by photogrammetry to produce coloured 3D meshes. These were split into multiple components, which were each varied smoothly to generate intermediate values. These intermediates were then recombined into a single 3D object, and finally printed using additive manufacturing. **b**–**g**, How each component of the phenotype (shape (**b**), pattern (**c**), size (**d**) and colour (**e**)) of the body, plus simplified representations of legs, wings and antennae (**f**)) is interpolated, and the resulting axis of mimetic accuracy (**g**).

fitness landscape of mimetic accuracy beyond the examples seen in nature.

To test key hypotheses about the existence of inaccurate Batesian mimics, we first tested the extent to which wild predators discriminate highly accurate, yet imperfect, mimics from their models. We then tested whether the presence of more than one model species affords increased protection to intermediate mimics (the multiple-models hypothesis[12]). We tested the relative signal salience[22] of shape, colour, pattern and size by varying those traits independently and testing which are under the strongest selection for accuracy. Finally, we tested the eye-of-the-beholder hypothesis[11] by comparing the responses of a range of insectivores towards the same set of mimetic stimuli.

## Discrimination ability

In the few tests of predator discrimination between real models and mimics, birds consistently distinguish images or specimens of wasps from any tested hoverfly, including some seemingly accurate Batesian mimics[23,24]. However, using real specimens or images can never reveal the possible protection of a hypothetical mimic even closer to the model phenotype. The degree to which the fundamental limits of predator perception and cognition constrains decisions to attack mimicry complexes is therefore uncertain.

We generated stimuli (Fig. 1) along three axes, with each axis using the common wasp *Vespula vulgaris* to represent the aversive model at one end point, denoted V100 (100% based on *V. vulgaris*). Each axis used a different fly (Diptera) taxon as the other end point: the non-mimic *Mesembrina meridiana* (M100), intermediate mimic *Syrphus ribesii* (S100) and accurate mimic *Chrysotoxum* spp.[3] (C100; Extended Data Table 1). We selected three intermediate points on each axis corresponding to equally spaced values of shape, colour, pattern and size. For example, S25/V75 indicates a stimulus of 25% based on *S. ribesii* and 75% based on *V. vulgaris*. To the human eye, the most accurate of these mimics appears considerably more like *V. vulgaris* than any existing hoverfly. The axis *M. meridiana* to *V. vulgaris*, viewable digitally in 3D, is provided in Supplementary Data 1.

The focal predators were wild, free-living great tits (*Parus major*) in Madingley Wood, Cambridge, UK. Great tits are generalist predators of Hymenoptera and Diptera[25]. We trained them to forage from feeding stations presenting arrays of small opaque dishes covered by openable lids that concealed a mealworm (*Tenebrio molitor*). We then fixed 3D stimuli to the lids to signal the reward status: half of the dishes displayed a non-mimetic fly stimulus (M100) and contained a mealworm, and half displayed a model wasp stimulus (V100) with no reward (Fig. 2a). The dishes were reset daily for a new session with a randomized arrangement of stimuli. Birds visited the feeding stations repeatedly during a session and could select among any unopened dishes at any time. We assumed that the birds attempted to maximize their rate of food consumption[26] to minimize opportunity costs and predation risk, starting with the dishes that they perceived most likely to be rewarding. We therefore estimated the level of protection of each stimulus according to how early or late in the sequence that dish was opened.

During the first day of training, the birds showed no bias towards either stimulus (47% for M100 and 53% for V100 among the first 15 dishes per feeder; binomial test, $P = 0.38$; $n = 45$), suggesting that previous encounters with real wasps or flies did not influence foraging choices (this was winter, when the birds had probably not encountered wasps for several months). After 3 weeks of training, the birds

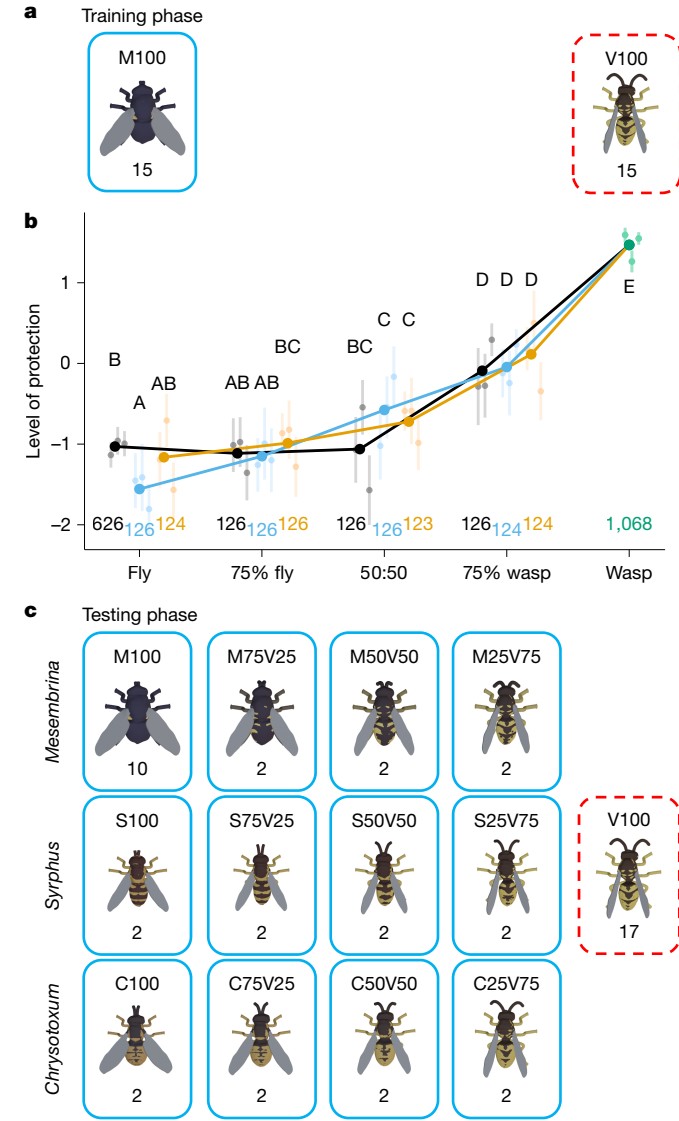

**Fig. 2 | Discrimination ability of great tits. a**, The design of the training phase. The solid blue border represents the rewarded stimulus, and the dashed red border represents no reward. The code above is a unique label for each stimulus type, with the letters indicating the species used as an end point and the numbers indicating the weighting. The numbers below indicate how many of that stimulus type appeared on a single feeding station, within an array of 30 dishes. **b**, The levels of protection received by different stimulus types resulting from great tit behaviour during the testing phase. Level of protection is the rank order of attack for that stimulus within a session, logit transformed. A higher level of protection indicates that a stimulus was attacked later in the sequence, or not at all. The bold points show the mean for all feeders; faint points show the means for individual feeders along with 95% confidence intervals based on the *t*-distribution. The black line shows the *Mesembrina* axis, the blue line shows the *Syrphus* axis and the orange line shows the *Chrysotoxum* axis. Capital letters indicate groupings that show no significant difference after a Tukey post hoc test (*P* > 0.05). Numbers give the sample size (number of dishes). **c**, Design of the testing phase. Models from three axes were presented together at each feeding station, starting from three different fly species and all ending at *V. vulgaris*. The total array size was 49 dishes. Note that M100 has a larger sample size than S100 and C100 so that the birds faced some rewarding stimuli familiar from the training phase.

consistently targeted M100 first and either opened V100 dishes last or not at all (M100 88% V100 12% among first 15 dishes per feeder across the final 3 days of training; binomial test, *P* < 0.0001; *n* = 135; Extended Data Fig. 1a).

We next presented the full range of stimuli from all three axes, with approximately one-third being unrewarding V100 dishes, one-fifth being rewarding M100 dishes and the remainder being rewarding mimic dishes sampled from the other axis points (Fig. 2c and Supplementary Video 1). Birds generalized immediately from their learned preference towards the fly dishes, targeting the least-accurate stimuli first and moving on to the more accurate mimics when no other options were available (Extended Data Fig. 1b). With further opportunity for learning, the birds increased their discrimination between mimics and models (Extended Data Fig. 1b,c). After around 10 days, the responses stabilized and the birds discriminated all mimics from models, targeting V100 dishes significantly later than all other stimulus types, and avoiding the 75% wasp-like mimics more than those 50% and below (*n* = 3,071 presentations; Fig. 2b and Extended Data Table 2).

All mimetic stimuli were prioritized over wasp stimuli, with a trend of increasingly wasp-like stimuli receiving greater protection (Fig. 2b), despite all mimics being associated with the same reward. This aligns with signal detection theory, which predicts that, when multiple prey types are available, predators should respond more cautiously towards perceived signals that are more likely to have originated from a model[27]. Moreover, the optimal response will be more cautious when models are more abundant, and/or when models carry a more severe cost[28]. For our birds, the consequence of incorrectly targeting a model is a small opportunity cost (if, for example, another bird gets to a mealworm first): ethical and practical constraints mean that higher costs would be difficult to implement. In nature, wasps potentially impose a more severe cost (a sting), but also bring potential nutritional rewards, so the resulting protection would depend on quantifying these outcomes.

In a separate validation experiment, we investigated the extent to which our printed stimuli were treated like real insects by the birds. We again trained them to discriminate flies from wasps, then substituted half of the printed stimuli with dead specimens of real flies and wasps. Birds targeted the printed flies first (the learned reward in the training phase), followed by the real flies (novel but rewarding), with printed and real wasps (both unrewarding) receiving the same level of protection (Extended Data Fig. 2). Thus, the birds distinguished between at least some of the printed stimuli and their real-life equivalents—which, considering our other results (Fig. 2), is unsurprising—but generalized from printed stimuli to real insects, indicating that they recognize a commonality between the two.

Predictions of optimal discrimination levels are only meaningful if predators possess the sensory and cognitive abilities to achieve those levels. Our birds could detect and remember very subtle differences between mimic and model appearance, and used these differences to select rewarding over unrewarding prey. These results align with theoretical predictions that, given enough experience[29], signal receivers should have the ability to discriminate between tiny differences. Our findings provide a crucial baseline for studies of mimicry, rejecting the argument that inaccurate mimics are already sufficiently accurate to be indistinguishable from their models, given favourable conditions for the predator. This implies that, if a bird chooses to avoid inaccurate mimics, it may be driven more by its motivation[28,30] than a lack of ability. Moreover, certain factors might increase the level of sensory or cognitive difficulty of a discrimination task, such as separation of prey in time or space preventing side-by-side comparison[31], prey movement[32] or the existence of multiple model species, as we explore next.

## Multiple models

Evidence from studies of Müllerian mimicry suggests that more complex prey communities cause predators to use broader generalization and a more conservative foraging strategy[16,33]. The addition of a second model species to a mimicry system may increase the fitness of intermediate Batesian mimics, although these 'jacks of all trades' may not necessarily outperform perfect mimics of either model[12]. Despite the

clear existence of model diversity in nature[34], almost all experimental studies of Batesian mimicry use a single model phenotype (but see refs. 35,36), and a key prediction from the multiple-models hypothesis remains untested regarding whether a mimic intermediate between two model phenotypes gains greater protection than one with an equivalent level of accuracy to one model, but further removed from the second.

To test this, we presented two distinct model (that is, unrewarding) stimuli to the birds: the common wasp *V. vulgaris* (V100) and a solitary wasp *Argogorytes mystaceus* (A100; Extended Data Table 1). The latter is also defended by a sting and displays black-and-yellow warning colours, but differs from the common wasp in appearance. We generated an axis including three stimuli intermediate between the two models (A75/V25, A50/V50 and A25/V75) and a further two stimuli at each end extrapolated along the same trajectory (A125/V−25, A150/V−50, A−25/V125 and A−50/V150). From the multiple-models hypothesis, the intermediate stimuli should receive greater protection than the extrapolated stimuli, despite equivalent similarity to a single model, due to their resemblance to the second model species. For example, A50/V50 and A−50/V150 are both 50 units from the model V100, but A50/V50 is much closer to the second model A100 (50 units) than A−50/V150 is (150 units). If there is any additive effect of mimicry to the two models, the intermediate A50/V50 should receive greater protection than the extrapolated A−50/V150.

We trained wild, free-living great tits and blue tits (*Cyanistes caeruleus*; the latter making a low proportion of foraging visits; Methods) to avoid model stimuli using the same approach as in the discrimination ability experiment. Six feeding stations were divided among two treatments: three with a single unrewarding model stimulus *V. vulgaris* V100 (1M treatment) and three including a second unrewarding model stimulus *A. mystaceus* A100 (2M treatment; Fig. 3a). The inclusion of the 1M treatment acted as a control, enabling us to compare directly whether the addition of a second model stimulus in 2M alters the protection received by any of the mimetic stimuli. Both treatments included a rewarding non-mimic fly stimulus *M. meridiana* (M100). Once birds were consistently targeting the M100 stimuli first (Extended Data Fig. 3a), we introduced the intermediate and extrapolated stimuli as rewarding (Batesian) mimics, alongside existing stimuli.

At the start of the testing phase, preference for M100 stimuli was strong, but this weakened over approximately 10 days before reaching an asymptote with lower levels (but not an absence) of discrimination among stimuli (Extended Data Fig. 3b,c). Protection received by the various mimetic stimuli declined with increasing distance from the nearest model (n = 5,987 presentations; distance term ΔAICc = −28.9, d.f. = 1; Fig. 3b and Extended Data Table 3). This pattern was similar across both treatments (treatment × distance term ΔAICc = 1.4, d.f. = 1) and there was no increase in protection for intermediate as opposed to extrapolated stimuli (intermediate × distance term ΔAICc = 3.2, d.f. = 2).

The early decline in predator selectivity suggests that the birds were no longer as motivated to discriminate among the numerous stimulus types introduced in the testing phase. This contrasts with the results from our discrimination ability experiment, in which the birds increased selectivity early in the testing phase as they improved their recognition of stimuli. Differences between responses towards M100 and V100 stimuli (common to both experiments) were stronger in the discrimination ability experiment (Fig. 2) compared with in both treatments of the multiple-models experiment (Fig. 3c), despite the latter offering no other fly-like stimuli. Assuming comparable predator populations, this may indicate that the birds found the mimics in the multiple-models experiment sufficiently challenging to discriminate that they were less motivated to try to do so[16]. One challenge may have been the inclusion of extrapolated mimics, meaning that the model(s) were no longer at one extreme of the axis. More generally, as both axis end points were based on Hymenoptera, these mimics may have been perceived, on average, as more wasp-like than the stimuli in the discrimination ability

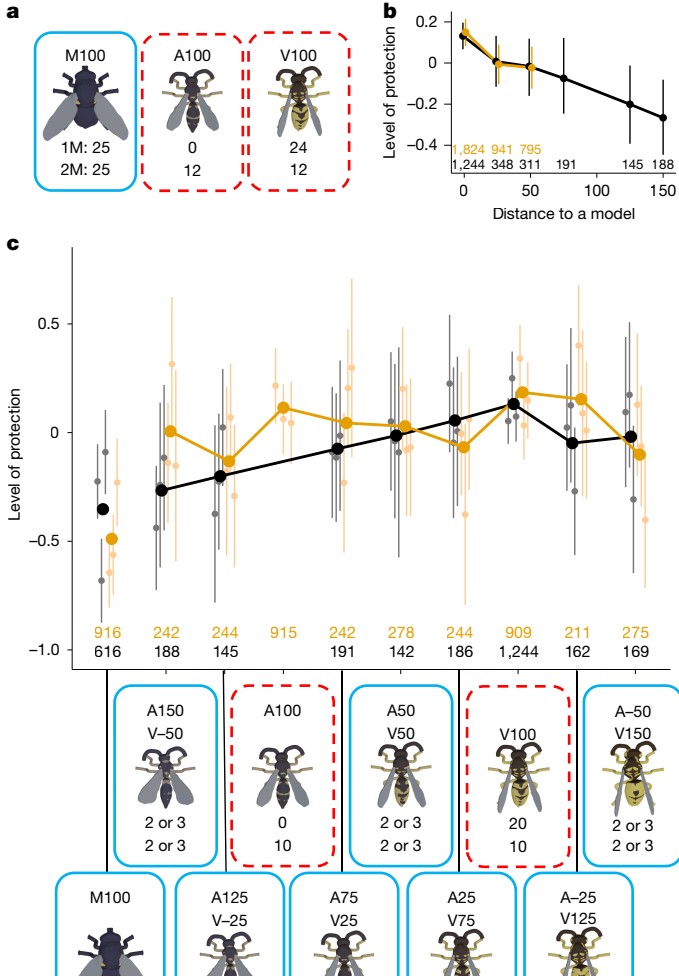

**Fig. 3 | Testing the multiple-models hypothesis. a,** The design of the training phase. The solid blue border represents rewarded stimulus, and the dashed red border represents no reward. The code above is a unique label for each stimulus type, with the letters indicating the species used as an end point and the numbers indicating the weighting (which is negative in some cases). The numbers below indicate how many of that stimulus type appeared on a single feeding station, for both one-model (1M, top) and two-model (2M, bottom) treatments. **b,** The levels of protection received by different stimulus types resulting from great tit behaviour during the testing phase. Stimuli are grouped according to phenotypic distance to the nearest model. Level of protection is the rank order of attack for that stimulus within a session, logit transformed. A higher level of protection indicates that a stimulus was attacked later in the sequence, or not at all. The 1M treatment is shown in black and 2M in orange. The points show the mean for all feeders along with the 95% confidence intervals based on the *t*-distribution. The numbers give the sample size (number of dishes). **c,** The same data as in **b,** but showing all stimuli separately. The bold points show the mean for all feeders and the faint points show the means for individual feeders along with the 95% confidence intervals based on the *t*-distribution.

experiment, including lower variation in certain key features associated with Hymenoptera such as the narrow waist and long antennae.

Despite the relatively low levels of selectivity, we still observed variation among stimuli in their level of protection. We found no evidence that a mimic gains extra protection by an intermediate resemblance to multiple models, compared with a mimic with equivalent accuracy to only a single model. However, our test is based on a single population of predators encountering all stimuli. In theory, multiple models could provide an additive benefit in cases where different predators

have learned to avoid different models, for example, due to allopatry or separate phenologies[9,12]. Further experiments incorporating geographical and/or temporal variation in mimetic communities would be required to test this.

Although we found no evidence for a selective advantage for jack-of-all-trades mimics, the addition of an extra model species is still important to the evolution of mimicry. In the two-model treatment, we see less variation among mimics in the level of protection received, because there are more phenotypes that are close to a model, and there are therefore more ways of achieving the same level of accuracy. In nature, aversive models frequently exist as part of a Müllerian mimicry ring of species with a shared warning signal[34]. In that context, there may be sizeable regions of phenotypic space in which a Batesian mimic would achieve similar levels of mimetic similarity to one member of the mimicry ring or another, and potentially experience relaxed selection for further improvements in accuracy.

## Trait salience

The above experiments varied all components of visual phenotype simultaneously, but some elements of appearance may have had more influence on predator behaviour than others[22,37]. Typically, colour is highly salient to birds, taking precedence over (overshadowing) other visual traits such as shape when choosing prey[19,22]. However, the informativeness of a trait is context-dependent and specific to the trait values under comparison[38]. If a salient mimetic trait has already evolved close resemblance to the model, predators may discriminate using other traits[37]. We must therefore consider trait values that are relevant to the study system when determining which traits are under the strongest selection. We sought to identify traits under the strongest selection for mimicry in the context of avian discrimination between wasps and flies. If some traits overshadow others, this could explain imperfection in otherwise conspicuous traits.

We generated experimental stimuli based on an axis from the non-mimetic fly *Tachina fera* to the wasp *V. vulgaris*. We varied four components of the appearance independently from one another, such that shape, colour, pattern and size could separately be fly-like (poor mimicry), wasp-like (perfect mimicry) or intermediate (good mimicry; equivalent to 50% in the discrimination ability experiment). We generated 31 mimetic phenotypes (Extended Data Table 4), with each mimic given combinations of poor and perfect traits, or good and perfect (but never poor and good, to limit the number of experimental subjects and presentations required).

To exert tighter control over the predator learning experience and facilitate the use of a larger number of stimulus types than in our wild-bird experiments, we conducted the experiment in the laboratory. We trained newly hatched chicks *Gallus gallus domesticus* in binary-choice trials to associate fly stimuli (poor in all traits) with a hidden mealworm reward and wasp stimuli (perfect in all traits) with no reward (Fig. 4a). Once the chicks showed a preference for the fly dish in at least 80% of presentations, we tested how they generalized their response to the various novel mimic (probe) stimuli in single presentations (Fig. 4b and Supplementary Video 2).

The chicks did not reject any presented prey, but did show significantly greater latency to attack the unrewarded stimuli (wasp: median 1.28 s (1.139 lower quartile, 1.48 upper quartile), $n = 545$ presentations, 30 chicks) compared with the rewarded stimuli (fly: 0.86 s (0.76 lower quartile, 1.04 upper quartile), $n = 544$ presentations, 30 chicks; stimulus term from linear mixed model: 0.43 s, $t = 24.5$, $P < 0.0001$). Even hesitations of fractions of a second, as seen here, could determine prey capture versus escape in a natural context[39], and therefore influence the selective pressures experienced[32]. The latency to attack was also significantly affected by stimulus type among the novel probe stimuli (Fig. 4c, Extended Data Fig. 4 and Supplementary Data 2). We tested both additive and nested models to predict chick response and found strong support for a positive association between the degree of colour mimicry and the latency to attack ($n = 910$ presentations, 30 chicks; colour appears as an additive effect in all five top-ranked models with $\Delta AICc < 2$; Supplementary Data 2). We also found some support for an influence of stimulus size on chick behaviour (size appears as an additive effect in three of the five top-ranked models with $\Delta AICc < 2$, and a further one of five as a nested effect when colour is good or perfect; Supplementary Data 2).

Our results reaffirm that colour should be under strong selection by birds for mimetic accuracy, being more salient than other visual traits[22]. Hoverfly and wasp colours are typically distinct enough to be theoretically discriminated by birds, but many are highly similar and could, under natural conditions, be indistinguishable[40]. In those cases, our results would predict selection to act on size as the next most salient visual trait. Our experiment demonstrates a strong propensity of chicks to respond to subtle size differences, even though our stimuli only differed by 2 mm at most (body lengths: wasp 12 mm, fly 14 mm). The ability to recognize Batesian mimics from their size is known in some garden birds[41], albeit involving larger differences. Pattern and shape appear to have weaker effects on predator behaviour, implying that these traits should experience relaxed selection[22], which would explain some elements of inaccurate appearance in mimics.

## Invertebrate predators

Many studies of Batesian mimicry, as with the experiments above, use birds as focal predators[16,18,22,23,41]. The eye-of-the-beholder hypothesis[11,23] states that mimics that appear inaccurate to one receiver might be perceived as more accurate to another. Consequently, the selective landscape for mimetic phenotypes depends on the suite of predators encountering a given mimic—the multiple-predators hypothesis[42]. Despite being important predators of many mimetic prey[43], and likely to attend to different aspects of a mimetic signal compared with vertebrates[42], invertebrates are under-represented as predators in mimicry studies. Invertebrates including praying mantises[44], jumping spiders[45] and crab spiders[46] can learn to avoid aposematic prey, and will generalize this avoidance to mimics, but other studies of invertebrate taxa have shown limited visual discrimination[47]. There is little evidence about how discerning these predators might be among mimics of varying accuracy (although see ref. 48) and none have compared the responses of multiple taxa to the same stimuli. Determining the extent to which perceptions of mimetic accuracy vary among invertebrates and differ from those of vertebrates is therefore of interest.

We assessed the ability of several invertebrates to discriminate between fly and wasp stimuli: praying mantises (Mantidae), jumping spiders (*Phidippus audax*) and crab spiders (*Synema globosum*). Owing to difficulties in training these predators to associate inedible stimuli with a separate reward, we trained them to associate the wasp stimuli V100 with a negative experience, leading to the same broad outcome of training: the spiders associated a non-mimetic fly (M100) with a more-positive outcome than V100. We then tested their response (Extended Data Table 6) towards stimuli from an axis running from M100 to V100 (as used in the discrimination ability experiment), with the aversive experience repeated in the case of V100 to reinforce the learning (Supplementary Video 3).

All three invertebrate taxa discriminated among the stimuli based on appearance (phenotype term in mantis models: $n = 40$ presentations, 8 individuals, $\Delta AICc = -36.4$, d.f. = 4 (Fig. 5a); jumping spider: $n = 57$ presentations, 9 individuals, $\Delta AICc = -26.8$, d.f. = 4 (Fig. 5b); crab spider: $n = 50$ presentations, 50 individuals, $\Delta AICc = -11.7$, d.f. = 4 (Fig. 5c); Extended Data Table 5). There was a sharp increase in protection from mantises between M75/V25 and M50/V50, after which point protection reached similar levels to the aversive V100 phenotype. Jumping spiders showed a similar pattern but with a more gradual increase in protection, only reaching wasp-like levels of protection at M25/V75.

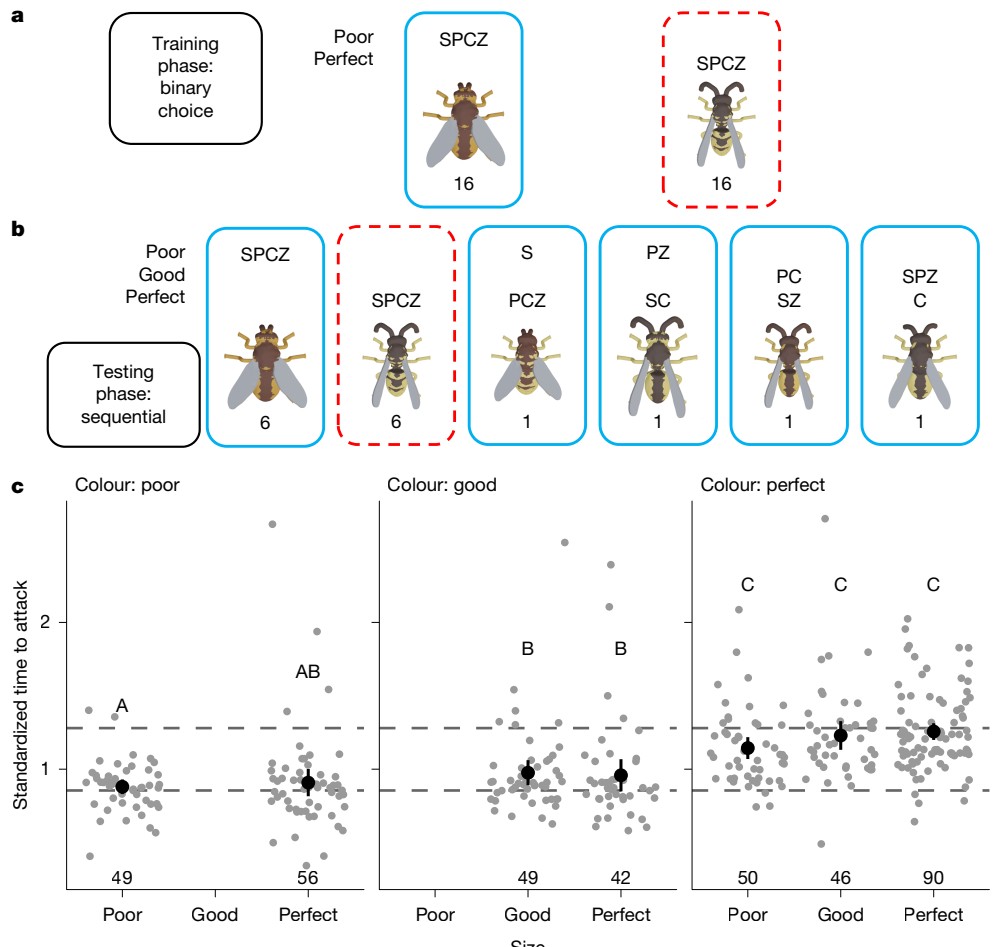

**Fig. 4 | Chick behavioural response to multiple traits. a**, Design of the training phase. Each trial consisted of 16 binary choices between *T. fera* stimuli with shape (S), pattern (P), colour (C) and size (Z) all poor, and *V. vulgaris* stimuli with shape, pattern, colour and size all perfect, as indicated by the codes above each stimulus image. The solid blue border represents the rewarded stimulus, and the dashed red border represents no reward. **b**, The design of the testing phase. Each trial consisted of 16 single presentations, including four probe stimuli, with the order randomized. Here we show one possible example of a selection of four probe stimuli, drawn from the options detailed in Extended Data Table 4. **c**, The latency to attack the stimuli, grouped by colour and size. Each column includes multiple combinations of shape and pattern traits. A plot with each trait combination shown separately is provided in Extended Data Fig. 4. The time to attack in seconds was standardized across trials by linear scaling such that values for fly and wasp presentations match the median values across all trials, shown as horizontal reference lines (wasp, top; fly, bottom). The bold points show the mean values and the vertical lines show the 95% confidence intervals based on the *t*-distribution. The faint points show the results of individual trials. The capital letters indicate groupings that show no significant difference after a Tukey post hoc test (*P* > 0.05). The sample sizes (number of presentations) are given at the bottom of the plot.

The trends for the crab spiders are less clear-cut, but suggest similar levels of protection for all stimuli of M50/V50 and above.

This demonstrates that, despite the overwhelming focus on vertebrates (especially birds) as predators in mimetic systems, invertebrates can also exert selective pressure for mimicry. Mimics sharing sufficient similarity with the models were afforded a similar level of protection to the models themselves, but the similarity required was considerably lower than that observed in the discrimination ability experiment using great tits. Moreover, among the three invertebrate taxa there were further differences in the level of mimicry required for protection.

Comparisons among such taxonomically diverse predators are inevitably limited by the need to tailor experiments to the behaviour and physiology of the predator in question. The mimetic phenotypes acceptable to a predator depend, among other factors, on the benefit of attacking a mimic and the cost of attacking a model[12], which are impossible to standardize fully across taxa that vary, for example, in nutritional needs and foraging strategy. Even focusing only on the invertebrates, which all received the same aversive stimulus, we see variation in the protection received by M50/V50, which was accurate enough to receive the same protection as a wasp from the mantises and

crab spiders, but not the jumping spiders (Fig. 5). A possible explanation is the varying visual abilities of the predators in question: they all use vision to hunt, but praying mantises have limited colour vision[49] in comparison to jumping spiders, which also have excellent acuity[50]. Even if all of the predators were to detect the same visual information about the stimuli, their varying behaviour could be explained by differences in foraging strategy, such as levels of risk-aversion. Regardless of the underlying mechanism, the identity of the predator is clearly a key factor in determining the strength of selection exerted on visual features of mimics[42].

## Discussion

Testing the responses of predators to 3D hoverfly-like stimuli enabled us to examine directly the protection received by various mimetic phenotypes. Our pipeline for generating intermediate and extrapolated phenotypes enabled us to evaluate areas of the adaptive landscape describing the protection received by Batesian mimics. Specifically, we have (1) generated highly accurate, but not perfect, mimics to show that such mimics still undergo selection for accuracy by a sufficiently

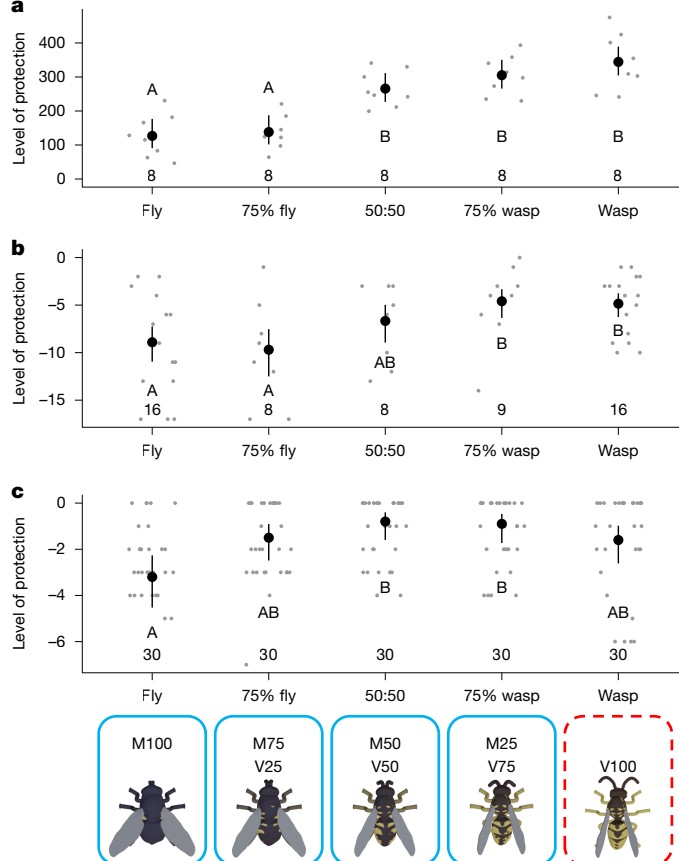

**Fig. 5 | Levels of protection received by different stimulus types resulting from invertebrate predators' behaviour. a**, The mantis (Mantidae spp.) level of protection is the latency to attack in seconds. **b**,**c**, The jumping spider (**b**; *P. audax*) and crab spider (**c**; *S. globosum*) level of protection is a count of aggressive behaviours towards the stimulus, subtracted from 0 to match the direction of response axes in other figures, with higher values indicating greater protection. The bold points show the mean values and the vertical lines show the 95% confidence intervals based on the *t*-distribution for the log-transformed positive data. Faint points show results of individual trials. The capital letters indicate groupings that show no significant difference after a Tukey post hoc test (*P* > 0.05). The numbers at the bottom of each panel give the sample size (number of presentations). The solid blue border represents the neutral stimulus, and the dashed red border represents the negative stimulus. The code above is a unique label for each stimulus type, with the letters indicating the species used as an end point and the numbers indicating the weighting.

motivated avian predator; (2) generated intermediates between two model species to show that jack-of-all-trades mimics receive no increase in protection compared with ones with similar accuracy to a single model; and (3) varied four visual traits independently to show that shape and pattern may be under weaker selection for accuracy than size and, in particular, colour.

Controlled testing of these hypotheses would not be possible using only real insect specimens. Our 3D-printed stimuli are good but not perfect visual replicas of real insects, inevitably being limited by available technology in aspects such as wing transparency and movement. Similar to most mimicry research[4,14,23,36], we focus here on explaining the visual components of mimicry. Other sensory modalities such as olfaction[51] and sound/vibration[52] may also contribute to predator discrimination, especially in the case of invertebrate predators[42], and could be explored in future work using a similar framework. Although our printed stimuli are not perceived by birds as identical to insects, birds will generalize from them to real specimens (Extended Data Fig. 2), demonstrating ecological relevance. Furthermore, both the realism of our stimuli

and the degree to which we can manipulate their traits represent a step change compared to previous studies using artificial prey[19,20].

We have conducted the most extensive comparison yet of invertebrate responses to the same mimetic stimuli, testing the eye-of-the-beholder hypothesis[11,23]. It is well known that varying visual systems can lead different predators to receive different information from the same signal[13], and here we have shown this variation can explain the persistence of inaccurate mimicry under selection from some predators. Among prey experiencing attacks from predators such as praying mantises, a wide range of only moderately accurate mimetic phenotypes will receive protection through their mimicry. Mimics attacked by more discerning predators will experience selection for greater accuracy whereas, in mimics exposed to multiple predators, selection will depend on the combination of different levels of discernment, and/or traits used to identify models and mimics by different predators[42].

Among the numerous proposed explanations for inaccurate mimicry, a key distinction lies between those suggesting an advantage to inaccurate mimics, and others predicting relaxed selection over a range of moderately accurate phenotypes[8]. We find no evidence in our experiments for a selective advantage to inaccurate mimics. Instead, we find that some traits, and prey of some predators, are likely to experience relaxed selection for visual mimicry.

The persistence of inaccurate mimicry in nature is a classic problem in evolutionary biology, notable for an abundance of theory much of which has been challenging to test experimentally. Our use of cutting-edge 3D technology enabled us to explore the selective pressures on mimetic adaptation in great depth, revealing the incredible discriminatory ability of an insectivorous bird, as well as how model community, trait salience and predator species limit the degree of discrimination in other contexts. In allowing such fine manipulation of visual phenotypes, our approach brings flexibility to the study of mimicry, and more widely to explore the adaptive landscapes for other complex morphological traits.

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

## Methods

### Production of artificial stimuli

**Overview.** To explore predator responses to realistic, but in some cases hypothetical, stimuli, we produced 3D-printed plastic insect replicas. Some stimuli were based on real insect specimens and were matched as closely as possible in shape, colour, pattern and size to the assigned insect. Other stimuli were produced by interpolating along a smooth gradient or axis running between two real specimens.

**Specimens.** Experimental stimuli were based on real wasp (Hymenoptera) and fly (Diptera) taxa chosen to represent different levels of mimetic accuracy (Extended Data Table 1). To generate intraspecific variation, in the wild-bird experiments, we used three different individuals from each taxon to produce separate stimuli.

Insect specimens were collected between June 2020 and August 2021 from various locations in England using a hand net. Specimens were euthanized by freezing at −18 °C for approximately 30 min. They were then pinned through the thorax and positioned into a natural-looking posture before drying for 6–24 h.

**Photogrammetry.** 3D digital images of the insect specimens were obtained by photogrammetry, using a protocol adapted from a previous study[53]. Specimens were suspended, with the anterior uppermost, on a motorized turntable (Genie Mini II; Manfrotto, Cassola, Italy), positioned against a white background and lit indirectly using two LED panel lights (22 W, 5600 K; Pixapro). They were photographed using a DSLR camera (Canon EOS 600D) and macro lens (Tamron SP 90 mm) with F20, 1/6 s exposure, ISO400. Each specimen was photographed from 36 different angles—three vertical camera positions at each of 12 equally spaced turntable orientations. Wings were removed and photographed separately (single photo at a perpendicular angle), because otherwise their positioning on the body prevented important details of the abdominal pattern and shape from being accurately reconstructed.

A 3D shape file (mesh) was built from the set of 36 photographs using the software 3DSOM[54], which uses the outline of the specimen in each photograph to carve out a 3D shape. The colour information from the photographs was then projected onto this 3D shape to give the corresponding colour pattern.

**3D image processing.** Except where noted, 3D images were edited using Blender[55]. Using the images obtained from photogrammetry as starting points, we constructed axes of similarity between pairs of real insects through 3D morphological space. We defined axes of phenotypic variation along which the traits of shape, colour, pattern and size varied smoothly from one image to the other, and generated phenotypes by picking either intermediate or (in the multiple-models experiment) extrapolated points along those axes. The four traits were varied in parallel with each other, except in the trait salience experiment, in which they were varied independently. Details of the specimen images used as axis end points are given under each experiment heading below.

Owing to difficulties in both processing and printing of thin and elongated structures, legs and antennae were removed digitally from the meshes, to be added back at a later step in more simplified form. Wings were treated in a similar manner, having already been removed from specimens before photogrammetry.

Shape deformations were carried out using the software Deformetrica[56], which uses control points based large deformation diffeomorphic metric mapping. A single simplified template was projected onto both end points, such that each retained its shape features but remapped onto new vertices that now had a direct one-to-one correspondence between the two meshes. We then calculated the deformation of 3D space required to transform one shape into the other and, using this, calculated intermediate shapes along the same axis.

Pattern manipulation was performed using custom scripts in R (v.4.3.0)[57]. Pattern data were mapped onto the reconstructed meshes for the two end points, and vertices were separated into two colour segments using $k$-means clustering ($k = 2$) of RGB colour values. A signed distance map was calculated for each end-point pattern, whereby all vertices were assigned a value being the shortest possible edge distance to a vertex of the opposing colour. We created new intermediate distance maps by taking weighted averages of the end-point distance maps, and then reverse-engineered them into binary colour patterns by assigning all positive vertices to one segment and negative vertices to the other.

Each segment was assigned a single RGB colour value calculated as the median of the original colour data from the vertices included in that segment. Colours for intermediate patterns were calculated as linear interpolations between the corresponding segments in the end points. Ultraviolet reflection was ignored because there is no evidence of such colour components in wasp or hoverfly patterns[40].

Owing to limited resolution at the printing stage, legs and antennae for all meshes were given the same standardized shape and a uniform colour. The shape was based around a cylinder, with diameter 0.6 mm for legs and 0.8 mm for antennae (thinner antennae were found to be too fragile after printing). In the case of legs, articulations were added to separate the coxa, femur, tibia and tarsus, and, for antennae, the cylinder was bent into a gentle curve. Colour was taken from whichever of the two body colour segments most closely matched the majority leg colour of real specimens. Antennal length was matched against distances measured from the original 3D digital image, with intermediates calculated by linear interpolation.

Wings were created with a flat shape, 0.4 mm thickness, based on the outline taken from photographs, which corresponded to shapes as they are typically seen when the insect is at rest. In contrast to Diptera, *V. vulgaris* and *A. mystaceus* have two pairs of wings but, at rest, the hindwings are hidden owing to overlap with the larger forewings (the latter being folded in the case of *V. vulgaris*). Wing shapes for intermediate meshes were calculated using the same deformation method as for the bodies. As our printing method was unable to recreate transparent materials, all wings were assigned a uniform colour value of 50% grey. This colour matched that of the bases to which the insects were attached (see below).

The various components (body, legs, antennae and wings) were combined digitally to produce a mesh of the whole insect and finally scaled to match the body length of the relevant end point, or a value calculated by linear interpolation for intermediates. A base was added to provide an attachment point for the object as a whole, as well as improving the structural integrity of the legs. This base was circular as viewed from above, with a narrow post extending up into the ventral side of the thorax.

An example axis (*M. meridiana* to *V. vulgaris*), viewable in 3D, is provided in Supplementary Data 1.

**Additive manufacturing.** We printed physical 3D representations of these digital insects on a HP Jet Fusion 580 machine using polyamide 12 powder (CB PA12) and colour cosmetic settings. Stimuli were printed at Matsuura Machinery for the discrimination ability and invertebrate predators experiments, and at the University of Nottingham for the rest. Stimuli were then given VaporFuse Surfacing treatment in a DyeMansion Powerfuse S, which created a less grainy, slightly glossier finish.

**Nomenclature.** We refer to stimuli in the text according to the initial letter of the genus of the axis end points, and the percentages by which each was weighted when creating any intermediate form. For example, C100 indicates a stimulus based 100% on *Chrysotoxum*, and M25/V75 indicates an intermediate with *M. meridiana* weighted by 25% and *V. vulgaris* weighted by 75%. In the multiple-models experiment, some stimuli were created by extrapolating beyond the range of the two end

points, using weighted averages greater than 100% or below 0%, for example, A150/V−50.

### Ethical approval

The Trait Salience experiment was approved by Newcastle University AWERB committee (project ID 966). Wild-bird experiments (discrimination ability and multiple models) were approved by AWERB committees at University of Nottingham (project ID 260) and University of Cambridge (ref. NR2022/60).

### Wild-bird experiments

**Field site and study organisms.** Fieldwork was conducted in Madingley Wood, Cambridgeshire, UK (52.217° N, 0.049° E), a deciduous woodland composed primarily of broadleaf hardwood trees. The wood has a resident population of great tits, some of which, as part of other projects, have been fitted with passive integrated transponder (PIT) tags. Tags of birds involved in this study were fitted between July 2018 and October 2022 under licence from the special methods of the BTO projects 1120 and 1121 held by HMR. Birds included both males and females and were a mix of ages from first-year juveniles upwards.

**Feeding stations.** Feeding stations were placed at intervals within the wood, positioned close to dense vegetation to provide cover for small birds, and separated from each other by at least 80 m. The feeding stations consisted of a 0.75 × 0.75 m wooden board on which a 7 × 7 array of 30 mm diameter Petri dishes was fixed. The board was placed on top of a 1.4 m wooden post and covered with a 0.75 × 0.75 × 0.75 m cage made from 7 mm square galvanized wire mesh. On one side of the cage, approximately 0.5 m above the bottom of the cage, a 30 mm entrance hole allowed small birds to enter past a data logger antenna. The antenna was linked to a data logger (Francis Scientific Instruments), which logged PIT tags of any tagged birds entering. A single horizontal perch ran across the cage at the level of the entrance, and a further six perches were placed approximately 100 mm above the surface of the board, running between rows of Petri dishes. A motion-sensitive camera trap (CY70, Ceyomur) was placed above the top of the cage pointing downwards, such that the cage entrance and all Petri dishes were in view.

An example video showing two great tit individuals interacting with a feeding station is provided in Supplementary Video 1.

**Discrimination ability and multiple-model experiments.** Two main experiments were conducted at this field site using similar methodologies, along with a third generalization test: the discrimination ability experiment ran from December 2021 to May 2022, multiple models from October 2022 to April 2023 and the generalization test from October to December 2023. These experiments differed in timings and the stimuli used as explained in the relevant sections below, and a few details relating to sample sizes as follows.

In the discrimination ability experiment, five feeding stations were used; two of those did not receive enough successful feeding events and were dropped from the study before the testing phase, leaving three feeders. In the multiple-models experiment, a sixth feeder was added and all were in use throughout the experiment. In the generalization test, four feeders were used, three of which had been used previously and one placed in a new location within the wood.

Ten tagged individual great tits during the testing phase of the discrimination ability experiment, and eight tagged individuals in the multiple-models experiment, made more than 80 visits each, including five individuals present in both experiments. An unknown number of untagged individuals also visited in both cases; trapping records indicate that approximately 71% of the population were tagged in November 2021 and 51% in January 2023. In the discrimination ability experiment, tagged birds directed most of their visits to a single feeder (median 90%, lower quartile 78%, upper quartile 95%). During the multiple-models experiment, fidelity to feeder was weaker (median 51%, lower quartile

42%, upper quartile 64%) but fidelity to treatment was high (median 81%, lower quartile 66%, upper quartile 92%). In the generalization test, only one tagged individual visited the feeders, with 68% of its visits to a single feeder. No tagging had been conducted that year, so many tagged birds had probably died or dispersed.

**Stimuli: discrimination ability experiment.** Stimuli were drawn from three axes, all ending at *V. vulgaris* and starting from fly taxa with varying levels of mimetic accuracy: *M. meridiana, S. ribesii* and *Chrysotoxum* (Extended Data Table 1). Each axis consisted of the two end points and three intermediates at 25%, 50% and 75% similarity to *V. vulgaris*.

In the training phase, we used 15 rewarding fly (M100) and 15 unrewarding wasp (V100) stimuli (with 19 dishes left unused). In the testing phase, we used 17 unrewarding V100 stimuli and 32 rewarding stimuli, the latter including 10 M100 stimuli as experienced in the training phase, as well as two of each of 11 new phenotypes. New phenotypes were three intermediates from the *M. meridiana* axis (M75/V25, M50/V50, M25/V75), *S. ribesii* (S100) and its three intermediates (S75/V25, S50/V50, S25/V75), and *Chrysotoxum* (C100) and its three intermediates (C75/V25, C50/V50, C25/V25).

**Stimuli: multiple-models experiment.** Stimuli were drawn from an axis running from *A. mystaceus* to *V. vulgaris*, representing two model species and related phenotypes. In addition to the end points, each axis included three intermediates (25%, 50% and 75%) and four extrapolations, two beyond each end point at distances equivalent to the 25% and 50% intermediates. A separate non-mimetic stimulus of *M. meridiana* M100 was used, with no intermediates.

Feeders were assigned to either one model (1M) or two model (2M) treatments, with treatments spatially grouped within the study site to reduce the chances of an individual bird that visited multiple feeders experiencing both treatments. In the training phase, we used 25 rewarding fly (M100) and 24 unrewarding wasp stimuli, the latter being either exclusively V100 (1M treatment) or 12 × V100 and 12 × A100 (2M treatment; Fig. 3a).

In the testing phase, we used 20 unrewarding wasp stimuli, either exclusively V100 (1M) or 10 × V100 and 10 × A100 (2M), and 29 rewarding stimuli of which 10 were M100 and the remaining 19 were drawn in equal numbers (with rounding) from the intermediate and extrapolated phenotypes of the *A. mystaceus*−*V. vulgaris* axis A150/V−50, A125/V−25, A75/V25, A50/V50, A25/V75, A−25/V125, A−50/V150 (Fig. 3c).

**Stimuli: generalization test.** Here we tested whether the birds would generalize their preference for flies over wasps, learned from the printed stimuli, to the real insects. In the training phase, we used 12 rewarding fly (M100) and 12 unrewarding wasp (V100) stimuli (with 25 dishes left unused). In the testing phase, we swapped half of the printed stimuli for dead, real specimens of the same fly and wasp species, glued to circular bases identical to those used for the printed stimuli. The testing phase was limited to 5 days to focus on the birds' initial responses to the real specimens and minimize their opportunity to refine their learning. The short duration also minimized damage and decay of the specimens.

**Habituation phase (wild birds).** Feeders were first provided with open Petri dishes which contained a single mealworm per dish, as well as peanuts placed on the board in between dishes (only provided during initial stages and when visitation rates were low, to encourage birds to visit). Food was refilled every 2–3 days, and the whole feeding apparatus was sterilized using 70% ethanol spray every 2 weeks. After 3 days, transparent lids were placed onto the Petri dishes so that mealworms were visible, but only accessible if the lids were opened. Over the course of four weeks, visiting great tits learned to open the lids by flipping them off using their beaks. Petri dishes and lids were then painted so that the contents were not visible until the lids were flipped. Great tits continued

to search for food by flipping off the lids to obtain the mealworms and, in most cases, all 49 mealworms had been consumed after 2 days. Other bird species and small mammals were seen visiting the feeding stations to feed on the peanuts, but rarely opened lids. In the multiple-models experiment, from 12,331 lids that were opened, 401 were by blue tits *C. caeruleus*, which were included in analysis, considering their close relatedness with great tits. Mice opened 59 lids which were excluded from analysis. Only great tits opened lids in the discrimination ability experiment and the generalization test.

**Training phase (wild birds).** After the habituation phase, a 3D-printed stimulus was attached to the lid of each Petri dish. To train the birds to avoid the wasp stimuli (V100 and, for multiple models 2M treatment, A100), no food was provided in the corresponding dishes, and mealworms were placed only in the fly dishes (M100). Every 1–2 days we began a new session by replacing all of the lids in a new configuration, randomized with respect to board position, and restocking the relevant dishes with mealworms. The training phase continued for 3 weeks for the discrimination ability experiment, and 4 weeks for the generalization test. The training phase of the multiple-models experiment continued for 6 weeks, which included a gap of 1 week (Extended Data Fig. 3a) when cold weather forced a pause in the experiment because heavy frost made the dishes unopenable.

**Testing phase (wild birds).** The testing phase followed the same methodology as the training phase, but introducing a wider range of stimuli in addition to those on which the birds had been trained (see the 'Stimuli: discrimination ability experiment' and 'Stimuli: multiple-models experiment' sections). All of the newly introduced stimuli were rewarded, representing mimics with varying levels of accuracy. This phase lasted 5 weeks (10 weeks for the multiple-models experiment).

## Trait salience experiment

**Study organisms and housing (chicks).** Domestic chicks (*G. g. domesticus*; P.D. Hook Hatcheries) were acquired immediately after hatching and housed in a laboratory at Newcastle University. Chicks (not sexed) were housed communally in two non-concurrent batches of 36 in a floor pen measuring approximately 2 m$^2$ with access to food (HPS Starter Crumb, Special Diets Services) and water ad libitum. The room was kept at 25 °C and under a 14 h–10 h light–dark cycle. The number of chicks was chosen with the aim of a sample size of 10–20 presentations per stimulus type, allowing for some exclusions due to failure to meet training criteria (see below).

**Experimental arena (chicks).** The experiments took place in an arena measuring 140 × 70 × 40 cm and divided into three sections of lengths 25, 90 and 25 cm, separated by mesh barriers such that each section was visible from the others. The first section formed a buddy area to house two buddy chicks (from a stock of eight, rotated every hour) during all sessions. Buddy chicks were never used for experimental testing, but instead ensured that experimental chicks were always able to see and hear conspecifics, to reduce stress. The largest section of the arena was the experimental area, which included a removable board on which grey opaque food dishes, with removable lids, were mounted. The final section was a holding area in which chicks were placed during 30 s gaps between presentations.

An example video showing a chick approaching stimuli in the experimental arena is provided in Supplementary Video 2.

**Stimuli (trait salience).** Stimuli were based on the non-mimic *T. fera* and the model *V. vulgaris*. Each of four traits—shape, colour, pattern and size—was varied independently to different levels of mimicry, being poor (matching *T. fera*), good (50% intermediate) or perfect (matching *V. vulgaris*). Stimuli were created in all possible combinations of poor and perfect traits, or good with perfect traits (but never poor and good

traits in the same stimulus), resulting in 31 different trait combinations (Extended Data Table 4).

**Habituation phase (trait salience).** On the first day after arrival in the laboratory, chicks received six 2 min trials in the experimental area, foraging from eight open dishes containing mealworms *T. molitor*. Chicks were first grouped in threes, then pairs, then individually (two trials each). Before the last three sessions on day one, and all of the following sessions, chicks were food-deprived for 60 min to ensure motivation to forage.

Over the course of the following 6 days, chicks received one trial each day during which they received 16 presentations of two dishes, each containing a mealworm. During a presentation, chicks were placed in the main arena and had up to 30 s to obtain a mealworm. Chicks were removed before being able to consume the second mealworm and placed in the holding area for 30 s in preparation for the next presentation. Each day, opaque lids were placed increasingly covering the dishes until the lids were fully on and the mealworm completely hidden, teaching chicks to lift off a lid to obtain a mealworm.

**Training phase (trait salience).** Chicks were each given a further series of trials during which they learned to discriminate fly from wasp stimuli through paired choices. Chicks were presented with the same two dishes as in habituation, but with one bearing a 3D-printed model of *T. fera* (fly, poor in every trait) and a mealworm inside, and the other with a model of *V. vulgaris* (wasp, perfect in every trait) and no reward. After the chick opened one of the two lids (or 30 s elapsed, whichever happened first), it was moved back into the holding area to prevent it accessing the other dish. The chicks then remained in the holding area for 30 s before the next presentation. Each day, the chicks received 1 trial of 16 presentations.

After 5 days of training the first batch of chicks, it was noted that some individuals showed a bias towards one of the two dish positions (left or right, not consistent across chicks), regardless of the stimulus. To reduce this stereotyped behaviour and encourage learning, we subsequently varied dish positions among presentations, placing dishes in two out of four possible positions along a line perpendicular to the chicks' starting position.

Trials continued until chicks chose the fly dish on at least 13 of the 16 presentations, which took 7–11 days. We excluded 12 chicks that did not reach this learning threshold from further testing and analysis. We note that, as a result, our conclusions apply only to the subset of the chicks involved in the testing phase. The presence of some individual predators which are less selective does not prevent the majority, which do discriminate among prey types, from exerting selective pressure on mimetic phenotypes.

**Testing phase (trait salience).** Chicks then received up to four further daily trials (some chicks that took longer to complete the training phase spent less time in the testing phase) testing their response to intermediate stimuli. The structure of trials was identical to the training phase, except that birds were given only one stimulus in each presentation. In each trial, chicks received six presentations of a Petri dish containing a mealworm and topped with the same fly stimuli used in training, six with no reward and topped with the wasp stimuli used in training and four further probe presentations. The probe stimuli were dishes containing a mealworm and topped with a novel insect, drawn at random from 31 possible trait combinations (Extended Data Table 4). Note that possible probe stimuli included one identical to the unrewarding wasp stimulus (perfect in all traits) but associated with a mealworm reward, so acting as a perfect Batesian mimic.

Chicks opened all dishes in the testing phase, without exception. The latency to attack was measured from the moment the chick was released into the arena to its first peck of the dish or lids. Given the speed at which chicks approached the dish (median, 1.1 s), timings were

taken from video recordings slowed to 0.3× speed using the BORIS software package[58] to improve accuracy. Experimenters were not blind to stimulus type during this process.

### Invertebrate predators experiment

**Study organisms and housing (invertebrates).** Praying mantises of three species (*Rhombodera kirbyi* ($n = 5$, fourth instar to adult), *Polyspilota aeruginosa* ($n = 1$, subadult) and *Pseudoxyops perpulchra* ($n = 2$, third instar), all unsexed; BugzUK and LDW bugs) and jumping spiders (adult male and female *P. audax* obtained from Jumping Spiders Web) were housed individually in transparent plastic boxes ($19 \times 13 \times 8$ cm) in a laboratory at University of Nottingham. The room was kept at 26 °C and under a 12 h–12 h light–dark cycle. They were fed crickets or mealworms twice weekly, with all trials conducted 30 h after feeding.

We collected crab spiders (*S. globosum*, adult male and female) that were sitting in wait for prey on flowers (where they hunt for pollinating insects) around the Quinta de São Pedro field research station (38.568° N, 9.193° W) and surrounding areas of Sobreda, Portugal. Individuals found with recently killed prey were not included. Spiders were kept at the Quinta de São Pedro research station, and individually housed in transparent plastic universal tubes. Spiders were kept, unfed, for 48 h until use, but note that the median time since the last meal will have been longer. The room was kept at 22 °C, with no artificial light–dark cycle.

**Experimental arena (invertebrates).** Mantis and jumping spider trials were performed inside an opaque plastic box ($19 \times 13 \times 8$ cm). A fishing line was fed through two small holes at either side of the box, with one end attached to a counterweight maintaining tension and the other end attached to a bobbin. The bobbin was spun by a motor, programmed with a microcontroller board (Arduino) to rotate in a randomized pattern (1–2 s clockwise or anticlockwise, 0–1 s pause, then repeated in the opposite direction). This moved stimuli left and right in rapid, jerky motions and encouraged striking[59]. Stimuli were suspended by a fine steel wire loop from the fishing line, allowing them to dangle and move in three dimensions.

Crab spider trials used a similar arrangement of equipment but with an arena that was larger ($69 \times 38 \times 41$ cm) and included the addition of a single purple milk thistle (*Galactites tomentosus*) to provide a perch for the spiders. The fishing line to which stimuli were attached entered through a hole in the lid of the arena as opposed to the side, causing stimuli to move vertically towards and away from the flower, as opposed to left and right.

Example video clips showing the different predators being presented with stimuli in their respective arenas are provided in Supplementary Video 3.

**Stimuli (invertebrates).** Stimuli were drawn from an axis running from *M. meridiana* (fly; M100) to *V. vulgaris* (wasp; V100) with three intermediates: M75/V25, M50/V50 and M25/V75. This axis matches one of the three axes used in the discrimination ability experiment. Stimuli were removed from their bases as the presentation method involved hanging down on a wire rather than resting on top of a lid.

**Training phase (invertebrates).** Praying mantises ($n = 8$) and jumping spiders ($n = 8$) each underwent six aversive conditioning trials on separate days. In the first trial, the stimulus was randomly allocated (M100 or V100) for each individual then, in subsequent trials, the stimulus was alternated. After being placed into the arena, animals were given 1 min to acclimatize before the stimulus was introduced. All mantises attacked the stimuli within 10 min, and were immediately punished after attacking wasp stimuli (V100) by being prodded firmly on the thorax with a separate wasp stimulus attached to the end of a thin metal rod. Subjects appeared to be appropriately threatened by this punishment, responding by moving away from the rod. Jumping spiders did not always attack, and were punished (in the same way as the mantises) at the end of trials involving a wasp stimulus, regardless of whether the spider had attacked the stimulus or not. Fly stimuli (M100) were associated with no reward or cost.

Training for crab spiders was performed using a condensed protocol as it was not possible to maintain the wild-caught spiders in the laboratory for long periods. The spiders ($n = 150$) did not undergo trials with presentations of wasp or fly stimuli, but simply received the punishment without any previous associated stimulus or behaviour. However, this still provided an opportunity to associate the negative experience with the wasp stimulus owing to its use in the 'punishment' process itself.

**Testing phase (invertebrates).** Praying mantises and jumping spiders received a further nine trials using the same procedures as the training phase. Five probe stimuli were presented consisting of each of the five points along the axis in random order, alternating with four reinforcement trials (two M100 and two V100). All attacks on (mantises) or encounters with (spiders) wasp stimuli were punished as before. Owing to restricted time in captivity, each crab spider was presented once with a single stimulus, selected from the five axis points at random.

As in the training, mantises attacked all stimuli, and the latency to attack was measured from when the motor was switched on to the mantis first striking the stimulus. Spiders rarely attacked the stimulus (*P. audax* 11% of trials, *S. globosum* 17% of trials); thus, using latency to attack as the primary measure of behaviour would provide poor resolution. They did, however, display a range of positive (such as orientation towards the stimulus, approach) and negative (for example, retreat, hide) behaviours in response to stimuli; a full list of the observed behaviours is shown in Extended Data Table 6. Instances of these behaviours were recorded over the full trial period (*P. audax*, 5 min; *S. globosum*, 3 min). Experimenters were not blinded to the stimulus type during the laboratory trial.

### Statistical analysis

All analysis was carried out in R (v.4.3.0)[57]. We used generalized linear models and generalized linear mixed models implemented in the package lme4 (ref. 60). In all cases, model fit was assessed visually for normality of residuals and homoscedasticity using residual plots. From a defined set of candidate models, the most parsimonious was selected based on lowest AICc values, with ties (a difference in AICc of less than two) broken by choosing the model with the fewest degrees of freedom[61].

**Wild-bird experiments.** Within each session and feeder, we determined the order of dishes being opened on the basis of video data, with any left unopened placed at the end of the sequence. Those coding the videos were not blinded to the stimulus identity during this process. We converted this sequence to a set of protection values from 0 to 1, corresponding to the first and last dishes of the sequence respectively. Thus, 0 can be considered to be the least protected as it is attacked first, and 1 the most protected as it is attacked last or not at all. These values were then logit-transformed using the formula $\log[(x + 0.01)/(1 - x + 0.01)]$ to occupy an unbounded scale, which improved normality of residuals. The 0.01 adjustment in this formula is to ensure that 0 and 1 transform to finite values[62].

For the discrimination ability experiment, initial preferences were assessed on the basis of bird behaviour in the first session of the training phase only. These preferences would have depended mostly or entirely on the subjects' innate sensory biases and learning from their experience in the wild and not on their experience with the experimental stimuli (although some learning may have taken place from the very first dish onwards). We used the explanatory variables reward (binary variable: mealworm or no mealworm, here corresponding to fly or wasp stimuli respectively) and feeder (categorical variable identifying the feeding station, here treated as fixed due to only having three levels).

We fitted the linear model preference ~ reward × feeder and compared it to all four nested submodels.

To highlight trends of stimulus selection as time progressed in the training phase, we fitted a nonlinear least squares (NLS) model to the levels of protection for the wasp stimulus (as there were only two stimulus types, the pattern for fly stimuli is simply an inversion of this). We used a sigmoid learning curve defined by the formula $\frac{a}{1+e^{-b\times(t-c)}}$ based on time $t$ measured as the number of sessions. This formula assumes that zero protection is received at time zero.

In the testing phase, we fitted separate curves to each phenotype, using an asymptotic curve with the formula $a + (b-a) \times e^{-e^{c} \times t}$ as, in contrast to in the training phase, there did not appear to be any initial warm-up period to the rate of learning. This approach enabled us empirically to parameterize the learning period according to the starting level of discrimination, rate of change and final level of discrimination, and therefore to identify a period of learning after which bird preferences were relatively stable. As a result, for our main analysis of the testing phase, we excluded the first 9 days of the testing phase while birds adapted to the new set of stimuli; from day 10 onwards, behavioural responses had reached within 10% of the asymptotic value according to the fitted learning curves. We used the explanatory variables feeder (as for training phase), plus axis (categorical variable for whether the stimuli were based on *Mesembrina*, *Syrphus* or *Chrysotoxum*), phenotype (the degree of similarity to the wasp, categorical to allow for a wide range of non-linear relationships) and edge (binary variable to indicate whether a dish was on the outer perimeter of the 49-dish array, included to improve the fit as we observed that birds preferred to open dishes along the perimeter of the cage; it was not tested for significance). We fitted the Gaussian linear model preference ~ (axis × phenotype + edge) × feeder and compared it to 32 nested submodels. This approach allowed the comparison of models with or without (1) an effect of phenotype, that is the degree of similarity to the wasp; (2) an effect of axis (*M. meridiana*, *S. ribesii* or *Chrysotoxum*), potentially interacting with phenotype; and (3) individual variation in behaviour, according to the interactions with the feeder term, since feeders represent largely separate sets of individual great tits. Tukey's post hoc comparisons were used to test for differences among different levels of the phenotype and axis variables.

For the multiple-models experiment, again we fitted learning curves using NLS, but found that patterns in the data conformed less closely to simple curve definitions. In the training phase, discrimination initially improved and then appeared to regress after a spell of cold weather (see the 'Training phase (wild birds)' section above), possibly owing to turnover of individuals. Asymptotic curves fitted to the whole training phase (formula as described for the discrimination ability experiment) did not converge but, when fitted just to the sessions after the experimental pause, converged for two of the three stimulus types. In the testing phase, asymptotic curves fitted using NLS did not converge on a solution for any stimuli. This is probably because most stimuli showed no clear trends with time, except for M100, which showed an increase in the levels of protection during the first 10 days. In the absence of fitted curves, we used the same cut-off as in the discrimination ability experiment, which matched our subjective assessment of the data trends, removing data from days 1–9 when modelling learned preferences.

We compared models representing several specific hypotheses related to great tit behaviour in the testing phase. We used the explanatory variables edge (as in the discrimination ability experiment), feeder (now fitted as a random effect as there were six feeders), treatment (categorical, one model or two models), distance to model (continuous variable measuring distance to the nearest model along the *Vespula*–*Argogorytes* axis of similarity; in the two-model treatment, A150/V−50, A50/V50 and A−50/V150 would all have the same distance (50 percentage points) from a model and would be predicted to elicit similar responses from the predators), intermediate (categorical variable indicating whether a stimulus lies between the two models along the axis of similarity; that is A75/V25, A50/V50 and A25/V75 are intermediate) and stimulus (categorical variable treating each position along the axis of similarity separately). All models used the Gaussian family and included fixed effects of edge and treatment and a random effect of feeder. H0: no effect of stimulus on bird preferences preference ~ edge + treatment + (1|feeder); H1: stimuli receive protection in inverse proportion to their distance from a model preference ~ distance_to_model + edge + treatment + (1|feeder). H2: as for H1, but intermediate mimics receive extra protection preference ~ distance_to_model * intermediate + edge + treatment + (1|feeder). H3: certain stimuli elicit avoidance behaviour in unique and unpredictable ways preference ~ stimulus + edge + treatment + (1|feeder). Each of H1–3 were also tested with the addition of an interaction between treatment and the stimulus-related term (H1 + distance_to_model:treatment; H2 + intermediate:treatment; H3 + stimulus:treatment).

**Trait salience experiment.** To standardize for variation in speed of behavioural responses of chicks among individual trials, within each trial, we compared the response to probe stimuli against values for the six fly and six wasp presentations. Within a trial, latency to attack was linearly scaled such that values for the response to fly and wasp presentations matched the median values from across all trials (0.855 and 1.28 s respectively). In 14 out of 105 trials, there was little (<0.1 s) or no delay in the mean response towards the wasp stimuli; these trials were excluded from further analysis.

We compared models representing several sets of hypotheses about the relative importance of the four phenotypic traits in influencing chick behaviour. We used the explanatory variables day (categorical variable for the number of days through the testing phase, allowing for changes in behaviour depending how many trials the chick has already completed), batch (categorical variable for which of two groups the chick belonged to, run on different dates), first_pres (binary factor indicating whether or not it was the first presentation of a trial, as we observed chicks to be slower on their first attempt of the trial), chick (random effect for individual ID), shape, colour, pattern and size (each represented by a three level factor of poor (fly-like), good (intermediate between fly and wasp) or perfect (wasp-like)), and interactions to represent overshadowing, where the overshadowed trait is ignored unless another trait, termed the main trait, is above a certain level of accuracy. All models used the Gaussian family and included fixed effects of day, batch and first_pres, and a random effect of chick. H0: no effect of stimulus on chick behaviour latency_to_attack ~ day + batch + first_pres + (1|chick). H1: each trait has a separate, additive effect on behaviour latency_to_attack ~ day + batch + first_pres + shape + colour + pattern + size + (1|chick). Nested submodels were also fit that excluded different combinations of the four trait terms. H2: one trait is assigned as overshadowing others, so that other traits are ignored unless the main trait is perfect (for example, latency_to_attack ~ day + batch + first_pres + colour + colour_perfect:shape + colour_perfect:pattern + colour_perfect:size + (1|chick)). The model was repeated with each of the four traits as the main trait, and nested submodels that excluded different combinations of the overshadowed traits were also fitted. H3: one trait is assigned as partially overshadowing others, so that other traits are ignored unless the main trait is good or perfect (for example, latency_to_attack ~ day + batch + first_pres + colour + colour_good_perfect:shape + colour_good_perfect:pattern + colour_good_perfect:size + (1|chick)), with variations as described for H2. H4: all trait combinations have their own unique effects on chick behaviour latency_to_attack ~ day + batch + first_pres + shape × colour × pattern × size + (1|chick).

**Invertebrate predators experiment.** The response variable for mantis behaviour was latency to attack, measured in seconds and modelled using a Gaussian family with log link. Jumping spiders and crab spiders rarely attacked the stimuli directly so instead, response was the number

of observations of positive behaviour towards the stimulus: display, approach and attack, and for jumping spiders, alert and orientation. These responses were modelled using a Poisson family with log link. Models included a fixed effect of phenotype (categorical variable for similarity to the wasp, as for discrimination ability above) and random effect of individual (except for the crab spiders, which had only one data point per individual). Tukey's post hoc comparisons were used to test for differences among different levels of the phenotype variable.

## Reporting summary

Further information on research design is available in the Nature Portfolio Reporting Summary linked to this article.

## Data availability

Data have been deposited at the NERC Environmental Information Data Centre and are available online: stimuli: 'Scanned 3D images and 3D printable images based on combinations of features of Diptera and Hymenoptera collected from the UK in 2021–22' (https://doi.org/10.5285/05169766-7355-4c3c-8ade-091db0583f9d); wild-bird experiments: 'Great tit behavioural responses to 3D-printed insect replicas, featuring combinations of traits from wasps and flies, in Madingley Wood, Cambridge, UK, 2021–2023' (https://doi.org/10.5285/a1c9b0cc-5585-49c5-a38f-fe05240edccf); trait salience: 'Chick behavioural responses to 3D-printed insect replicas, featuring combinations of traits from wasps and flies' (https://doi.org/10.5285/45391184-603e-4284-bb3c-9c8c6bf856ab); invertebrate predators: 'Invertebrate behavioural responses to 3D-printed insect replicas, featuring combinations of traits from wasps and flies, in laboratory trials' (https://doi.org/10.5285/ee7ba05a-449b-466e-840c-8de1d3f1d4d1). Source data are provided with this paper.

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

**Acknowledgements** We thank P. Wilderspin and the staff at the University Farm & Rural Estate (University of Cambridge) for permission to conduct fieldwork in Madingley Wood; T. Fulford, C. Thorne, J. Beaver and the members of the Madingley ringing group for PIT tagging great tits at Madingley; M. Waddle and the Comparative Biology Centre staff for technical assistance in chick husbandry at Newcastle University; D. Starkey for pilot work with jumping spiders and mantises; B. Richter for permission to conduct crab spider experiments at the Quinta de São Pedro field centre; L. Baker for coding of video data; and P. Harris for sharing their HP Jet Fusion expertise. The project was funded by a NERC standard grant (NE/S000623/1), with additional funding from the University of Nottingham, Leverhulme Early Career Fellowship (ECF-2018-700) to G.L.D. and the Max Planck Society and Royal Society (RG110122) to H.M.R.

**Author contributions** Conceptualization: C.H.T., F.G. and T.R. Methodology: C.H.T., D.J.G.W., J. Skelhorn, J.R.D., C.B., G.L.D., H.M.R., M.E., R.G., F.G. and T.R. Formal analysis: C.H.T. Investigation (wild-bird experiments): C.H.T., D.B., S.B., A.C., K.C., J.R.D., S.R.G., E.O., A.L.P. and T.R. Investigation (trait salience experiment): C.H.T. and D.J.G.W. Investigation (invertebrate predators experiment): J.J.D., T.D., H.J.J., R.L., H.R., J.R., J. Sains and F.S. Data curation: C.H.T. Writing—original draft: C.H.T. Writing—review and editing: all of the authors. Visualization: C.H.T., H.J.J. and J. Sains. Supervision: J. Skelhorn, F.G. and T.R. Project administration: C.H.T., J. Skelhorn and T.R. Funding acquisition: C.H.T., J. Skelhorn, C.B., R.G., F.G. and T.R.

**Competing interests** The authors declare no competing interests.

**Additional information**
**Correspondence and requests for materials** should be addressed to Christopher H. Taylor.

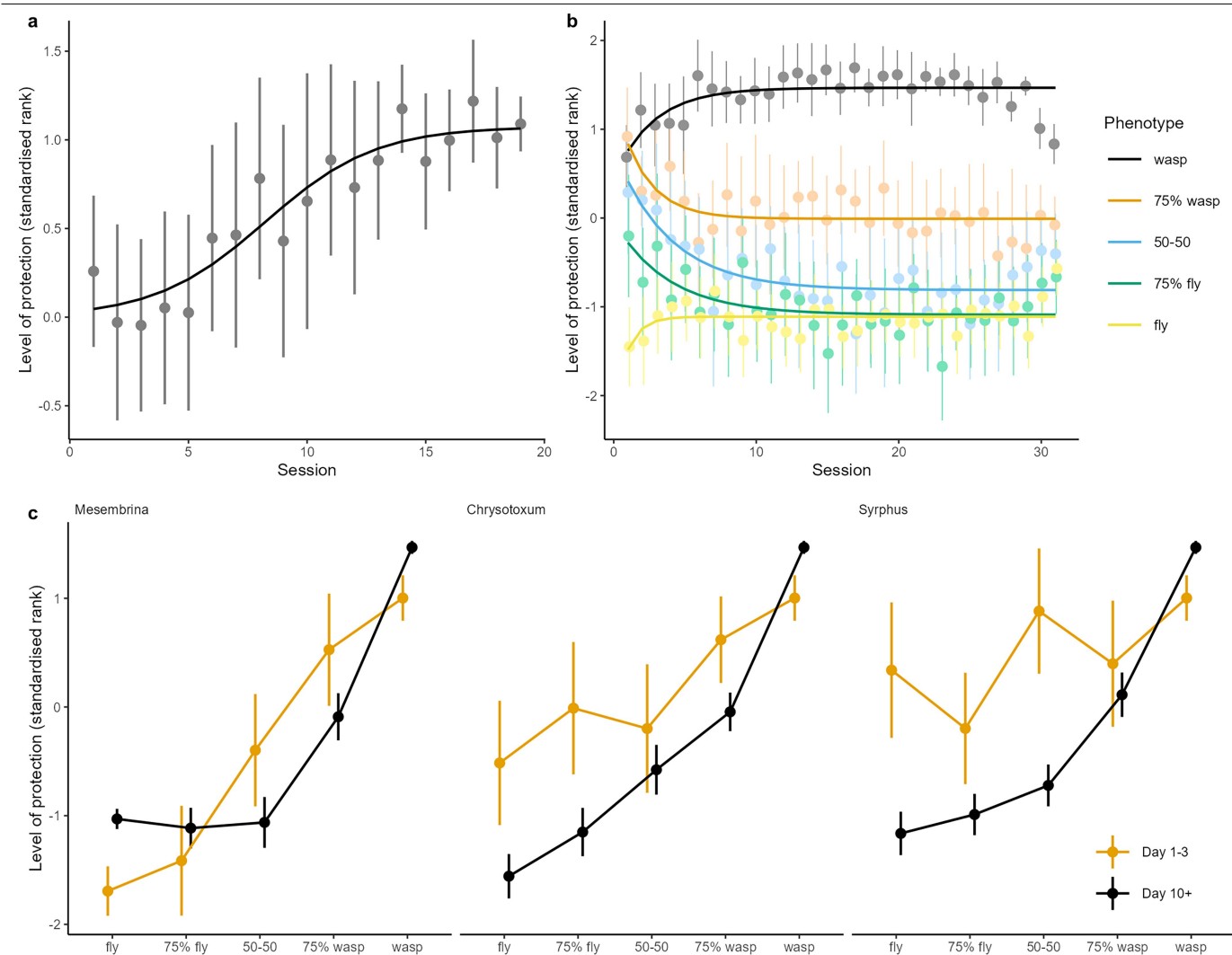

**Extended Data Fig. 1 | Changes in preference over time in the Discrimination Ability experiment.** "Level of protection" is the rank order of attack for that stimulus within a session, logit transformed. Higher level of protection indicates that a stimulus was attacked later in the sequence, or not at all. Points show mean and vertical bars show 95% confidence intervals based on the t-distribution. **a** Training phase. Data show wasp stimuli only; fly data are an almost exact inversion of the data shown. Line shows a sigmoidal curve fitted to the data. N = 828. **b** Test phase, trends with time. Asymptotic curves fitted to each phenotype. N = 1295 dishes (fly), 558 (75% fly), 553 (50-50), 552 (75% wasp), 1565 (wasp). **c** Test phase, comparing initial (yellow, sessions 1-3) and asymptotic (black, session 10 onwards, after fitted response reaches within 10% of the asymptote) preferences. Mesembrina axis, N = 191 dishes (initial), 1004 (asymptotic). Chrysotoxum axis, N = 96 (initial), 502 (asymptotic). Syrphus axis, N = 95 (initial), 497 (asymptotic). For images of the stimuli, see Fig. 2 in main text.

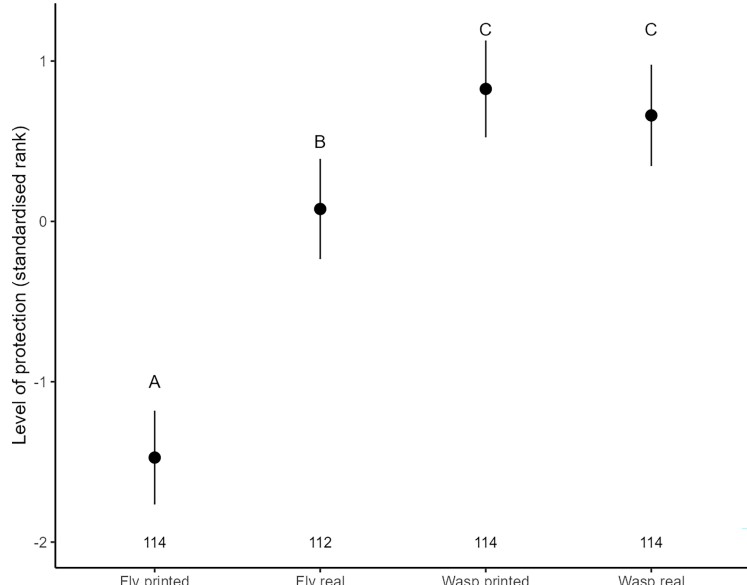

**Extended Data Fig. 2 | Validation experiment testing phase.** Points show mean and vertical bars show 95% confidence intervals based on the t-distribution. Capital letters indicate groupings which show no significant difference in a Tukey post-hoc test (p > 0.05). Sample sizes (number of dishes) are shown at the base of the plot.

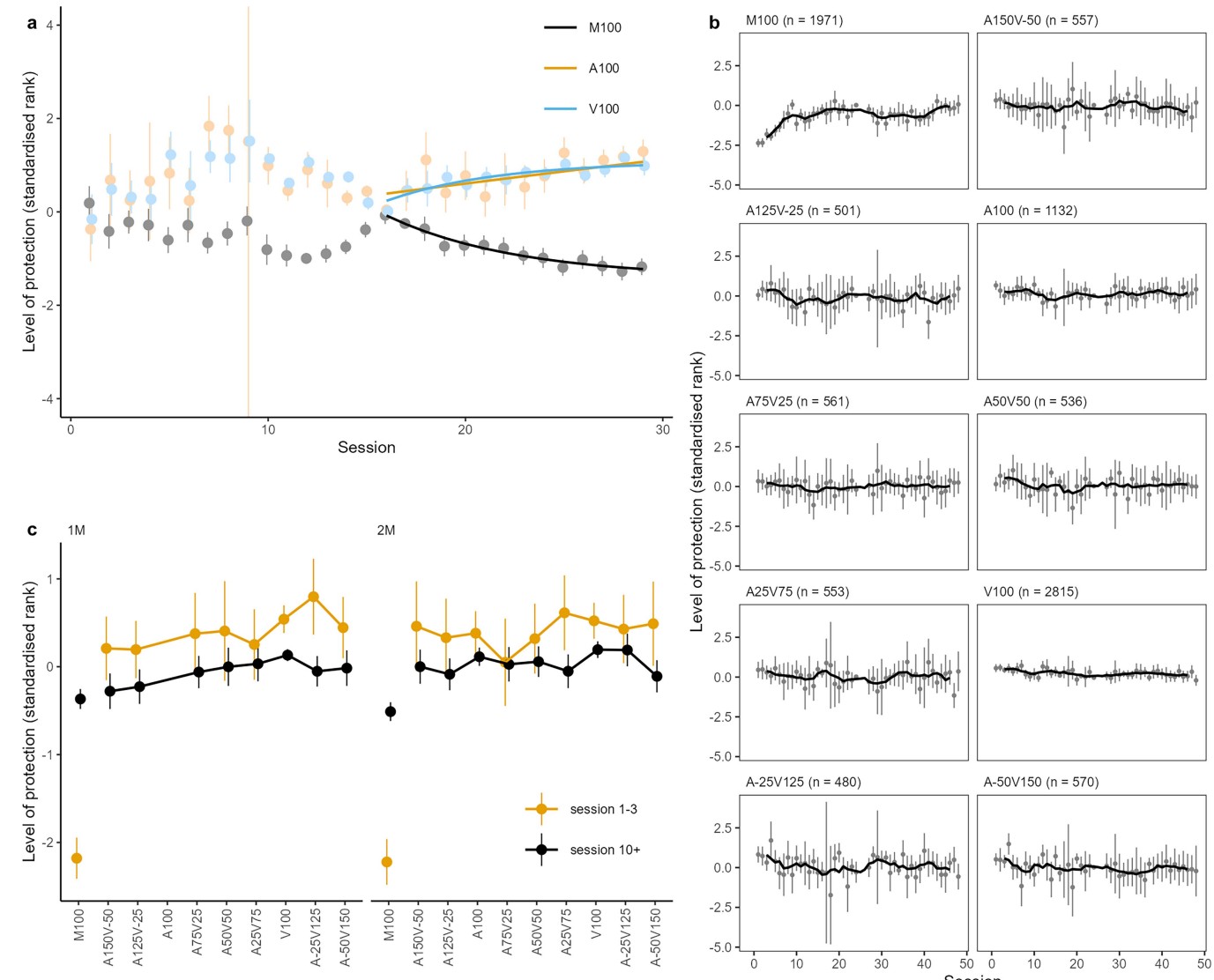

**Extended Data Fig. 3 | Changes in preference over time in the Multiple Models experiment.** Points show mean and vertical bars show 95% confidence intervals based on the t-distribution. **a** Training phase. Lines show an asymptotic curve fitted to the data from session 16 onwards; the curve for A100 did not converge but is shown for illustration. Session 16 was the first session after a period of one week when no birds opened dishes. Curves fitted to the full time period all failed to converge. A100 session 9 had a small sample size (3) and as a result has very wide confidence intervals (−5.3, 8.3) that are not shown in full for clarity of the rest of the plot. N = 3470 dishes (M100), 769 (A100) and 2227 (V100). **b** Test phase, trends with time. Asymptotic curves fitted to these data, using the same method as the training phase, failed to converge. Instead, trend lines are a moving average across five sessions (centred on the third session). Sample sizes shown above each plot. **c** Test phase, comparing initial (yellow, session 1-3) and asymptotic (black, session 10 onwards, chosen to match Discrimination Experiment) preferences. N = 437 dishes (initial, 1 M), 384 (initial, 2 M), 2902 (asymptotic, 1 M), 4334 (asymptotic, 2 M).

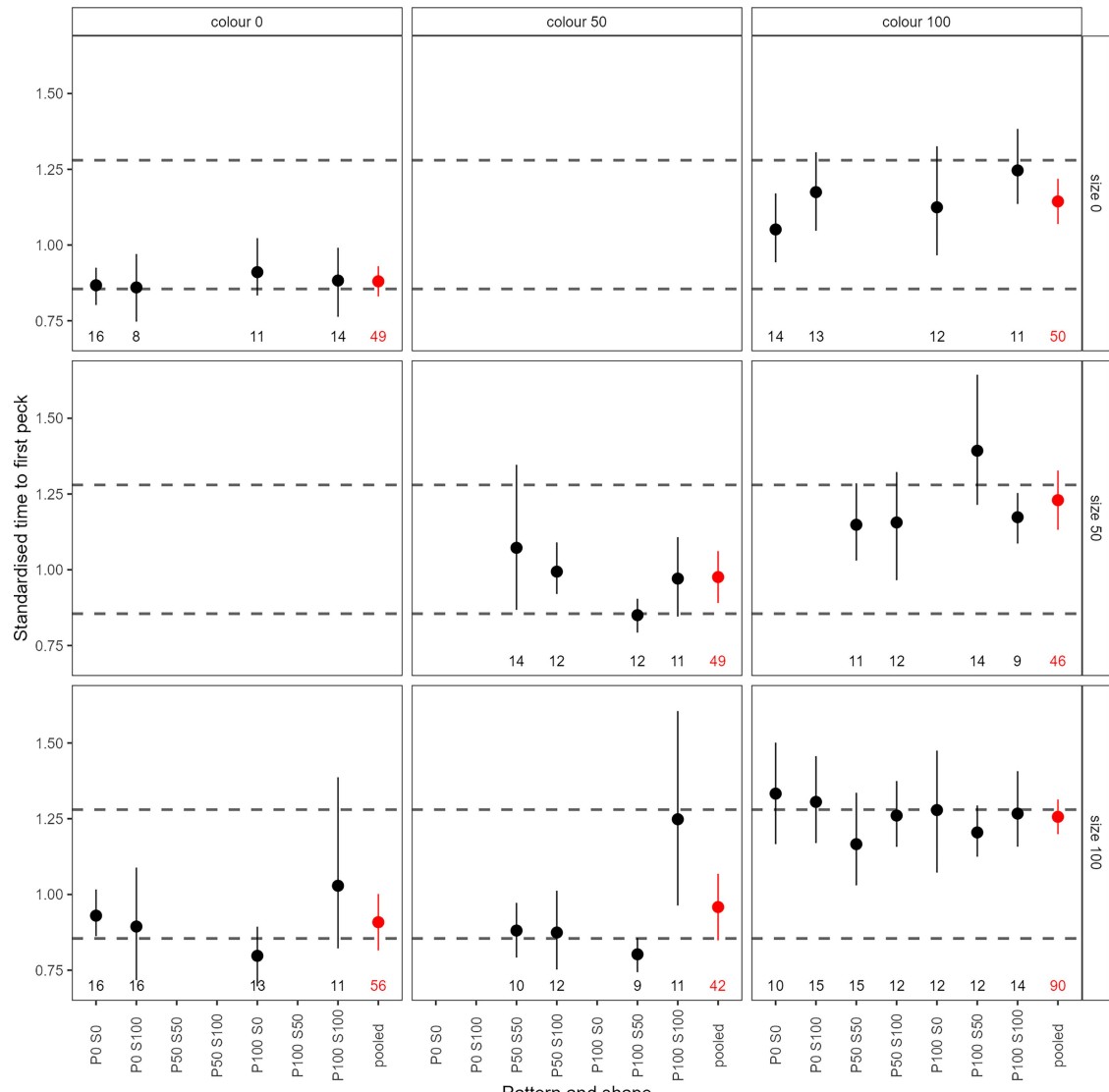

**Extended Data Fig. 4 | Chick latency to attack mimetic stimuli.** Levels of accuracy of mimetic traits are coded as 0 (fly-like/poor), 50 (intermediate/good) and 100 (wasp-like/perfect) for each of shape, pattern, colour and size. Each panel shows a certain combination of colour and size traits, and within a panel, black points show certain combinations of pattern (P) and shape (S), and red points show data pooled across all values for pattern and shape (as shown in Fig. 4, main text). Time to attack (seconds) has been standardized across trials by linear scaling such that values for fly and wasp presentations match the median values across all trials, shown as horizontal reference lines (wasp upper, fly lower). Points show mean and vertical bars show 95% confidence intervals based on the t-distribution. Sample sizes (number of presentations) are shown at the base of each plot.

**Extended Data Table 1 | Species on which 3D stimuli were based**

| Species | Order, Family | Role in mimicry | Reason for choice |
|---|---|---|---|
| *Vespula vulgaris* | Hymenoptera: Vespidae | Model | Very common model species known to be avoided by predators such as birds |
| *Argogorytes mystaceus* | Hymenoptera: Crabronidae | Model | Solitary wasp with distinct appearance to *V. vulgaris*, also armed with a sting |
| *Mesembrina meridiana* | Diptera: Muscidae | Non-mimic | Black and yellow like *V. vulgaris* but in a very different pattern, not considered mimetic |
| *Tachina fera* | Diptera: Tachinidae | Non-mimic | Common non-mimic fly species that is distinct from *V. vulgaris* in all four focal traits (colour, pattern, shape and size). |
| *Syrphus ribesii* | Diptera: Syrphidae | Batesian mimic | Considered a mimic of *V. vulgaris*, or possibly a generalized wasp mimic, due to black and yellow striped abdomen. |
| *Chrysotoxum arcuatum* and *C. verralli* collectively referred to as Chrysotoxum | Diptera: Syrphidae | Batesian mimic | Subjectively some of the most wasp-like hoverflies found in the UK. Two species grouped together here due to scarcity of specimens and being highly similar in appearance |

**Extended Data Table 2 | Comparison of models fitted to data from the testing phase of the discrimination ability experiment**

| Model rank | df | Log Likelihood | AICc | dAICc | additional terms* |
|---|---|---|---|---|---|
| 1 | 43 | -4602.2 | 9291.6 | 0.0 | a + f + p + a:f + a:p + e:f + f:p + a:f:p |
| 2 | 25 | -4625.4 | 9301.2 | 9.6 | a + f + p + a:f + a:p + e:f |
| 3 | 31 | -4621.8 | 9306.3 | 14.7 | a + f + p + a:f + a:p + e:f + f:p |
| 4 | 41 | -4618.9 | 9321.0 | 29.4 | a + f + p + a:f + a:p + f:p + a:f:p |
| 5 | 27 | -4633.5 | 9321.6 | 30.0 | a + f + p + a:p + e:f + f:p |
| 6 | 19 | -4643.2 | 9324.7 | 33.1 | a + f + p + a:f + e:f |
| 7 | 25 | -4639.7 | 9329.8 | 38.2 | a + f + p + a:f + e:f + f:p |
| 8 | 23 | -4642.7 | 9331.9 | 40.3 | a + f + p + a:f + a:p |
| 9 | 29 | -4638.8 | 9336.2 | 44.6 | a + f + p + a:f + a:p + f:p |
| 10 | 19 | -4652.7 | 9343.6 | 52.0 | a + f + p + a:p + e:f |
| 11 | 21 | -4651.3 | 9344.9 | 53.3 | a + f + p + e:f + f:p |
| 12 | 19 | -4653.4 | 9345.1 | 53.5 | f + p + e:f + f:p |
| 13 | 25 | -4650.1 | 9350.7 | 59.1 | a + f + p + a:p + f:p |
| 14 | 17 | -4661.0 | 9356.3 | 64.7 | a + f + p + a:f |
| 15 | 23 | -4657.1 | 9360.7 | 69.1 | a + f + p + a:f + f:p |
| 16 | 13 | -4670.2 | 9366.5 | 74.9 | a + f + p + e:f |
| 17 | 11 | -4672.3 | 9366.7 | 75.1 | f + p + e:f |
| 18 | 15 | -4670.3 | 9370.8 | 79.2 | a + p + a:p |
| 19 | 17 | -4670.0 | 9374.2 | 82.6 | a + f + p + a:p |
| 20 | 17 | -4670.2 | 9374.6 | 83.0 | f + p + f:p |
| 21 | 19 | -4668.4 | 9375.0 | 83.4 | a + f + p + f:p |
| 22 | 7 | -4690.1 | 9394.3 | 102.7 | p |
| 23 | 9 | -4688.3 | 9394.6 | 103.1 | a + p |
| 24 | 9 | -4689.8 | 9397.7 | 106.1 | f + p |
| 25 | 11 | -4688.0 | 9398.1 | 106.5 | a + f + p |
| 26 | 16 | -4779.5 | 9591.1 | 299.5 | a + f + a:f + e:f |
| 27 | 14 | -4795.8 | 9619.8 | 328.2 | a + f + a:f |
| 28 | 10 | -4804.3 | 9628.6 | 337.0 | a + f + e:f |
| 29 | 6 | -4821.0 | 9654.0 | 362.4 | a |
| 30 | 8 | -4820.7 | 9657.4 | 365.8 | a + f |
| 31 | 7 | -5792.6 | 11599.2 | 2307.6 | f + e:f |
| 32 | 3 | -5806.4 | 11618.8 | 2327.2 | |
| 33 | 5 | -5806.3 | 11622.5 | 2331.0 | f |

*All models are based on the initial formula (level of protection ~ edge), with the addition of the listed terms. Abbreviated terms: a = axis, e = edge, f = feeder, and p = phenotype.

**Extended Data Table 3 | Comparison of models fitted to data from the multiple-models experiment**

| Model rank | df | Log Likelihood | AICc | dAICc | additional terms* |
|---|---|---|---|---|---|
| 1 | 6 | -10460.8 | 20933.7 | 0.0 | distance |
| 2 | 7 | -10460.5 | 20935.0 | 1.4 | distance + distance:treatment |
| 3 | 8 | -10460.4 | 20936.9 | 3.2 | distance + intermediate + distance:intermediate |
| 4 | 9 | -10460.4 | 20938.9 | 5.2 | distance + intermediate + distance:intermediate + distance:treatment |
| 5 | 13 | -10458.4 | 20942.8 | 9.1 | stimulus (factor) |
| 6 | 20 | -10455.0 | 20950.2 | 16.6 | stimulus*treatment |
| 7 | 5 | -10476.3 | 20962.6 | 28.9 | - |

*All models are based on the initial formula (level of protection ~ edge + treatment + 1|feeder), with the addition of the listed terms.

**Extended Data Table 4 | Trait combinations used in the trait salience experiment**

| | | Shape: poor | | Shape: good | | Shape: perfect | | |
| --- | --- | --- | --- | --- | --- | --- | --- | --- |
| | | Pattern: poor | Pattern: perfect | Pattern: good | Pattern: perfect | Pattern: poor | Pattern: good | Pattern: perfect |
| Colour: poor | Size: poor |  |  | | |  | |  |
| | Size: perfect |  |  | | |  | |  |
| Colour: good | Size: good | | |  |  | |  |  |
| | Size: perfect | | |  |  | |  |  |
| Colour: perfect | Size: poor |  |  | | |  | |  |
| | Size: good | | |  |  | |  |  |
| | Size: perfect |  |  |  |  |  |  |  |

**Extended Data Table 5 | Comparison of models fitted to data from the multiple predators experiment**

| Model rank | Predator | df | Log Likelihood | AICc | dAICc | Formula right hand side |
|:---:|:---:|:---:|:---:|:---:|:---:|:---|
| 1 | Crab spider | 5 | -87.2 | 185.7 | 0.0 | stimulus |
| 2 | Crab spider | 1 | -97.7 | 197.4 | 11.7 | 1 |
| 1 | Jumping spider | 6 | -174.5 | 362.6 | 0.0 | stimulus + (1 \| id) |
| 2 | Jumping spider | 2 | -192.6 | 389.4 | 26.8 | (1 \| id) |
| 1 | Mantis | 7 | -220.7 | 458.8 | 0.0 | stimulus + (1 \| id) |
| 2 | Mantis | 3 | -244.3 | 495.2 | 36.4 | (1 \| id) |

**Extended Data Table 6 | Behaviours performed by *P. audax* and *S. globosum* in trials within the multiple predators experiment**

| Behaviour | Observed for | Description | Type |
|---|---|---|---|
| Bungee | *S. globosum* | Jumping off the flower and hanging from a line of silk | negative |
| Hide | *S. globosum* | Moving away from and out of sight of the model | negative |
| Retreat | *S. globosum* | Moving away from the model | negative |
| Orientation | *P. audax* | Orienting to face the model | positive |
| Alert | *P. audax* | Watching/tracking movements of the model | positive |
| Display | Both | Raising one or both of front legs towards the model | positive |
| Approach | Both | Moving towards the model | positive |
| Attack | Both | Jumping on or at, and/or biting, the model | positive |

# Reporting Summary

## Statistics

For all statistical analyses, confirm that the following items are present in the figure legend, table legend, main text, or Methods section.

| n/a | Confirmed | |
|---|---|---|
| ☐ | ☒ | The exact sample size (*n*) for each experimental group/condition, given as a discrete number and unit of measurement |
| ☐ | ☒ | A statement on whether measurements were taken from distinct samples or whether the same sample was measured repeatedly |
| ☐ | ☒ | The statistical test(s) used AND whether they are one- or two-sided<br>*Only common tests should be described solely by name; describe more complex techniques in the Methods section.* |
| ☐ | ☒ | A description of all covariates tested |
| ☐ | ☒ | A description of any assumptions or corrections, such as tests of normality and adjustment for multiple comparisons |
| ☐ | ☒ | A full description of the statistical parameters including central tendency (e.g. means) or other basic estimates (e.g. regression coefficient) AND variation (e.g. standard deviation) or associated estimates of uncertainty (e.g. confidence intervals) |
| ☐ | ☒ | For null hypothesis testing, the test statistic (e.g. *F*, *t*, *r*) with confidence intervals, effect sizes, degrees of freedom and *P* value noted<br>*Give P values as exact values whenever suitable.* |
| ☒ | ☐ | For Bayesian analysis, information on the choice of priors and Markov chain Monte Carlo settings |
| ☒ | ☐ | For hierarchical and complex designs, identification of the appropriate level for tests and full reporting of outcomes |
| ☒ | ☐ | Estimates of effect sizes (e.g. Cohen's *d*, Pearson's *r*), indicating how they were calculated |

*Our web collection on statistics for biologists contains articles on many of the points above.*

## Software and code

Policy information about availability of computer code

| Data collection | 3DSOM Pro v5<br>Deformetrica 4.3.0<br>Blender 2.9 |
|---|---|
| Data analysis | BORIS 8.19.3<br>R v4.3.0 with packages tidyverse 2.0.0, lme4 1.1, nlme 3.1, emmeans 1.10, and MuMIn 1.47 |

For manuscripts utilizing custom algorithms or software that are central to the research but not yet described in published literature, software must be made available to editors and reviewers. We strongly encourage code deposition in a community repository (e.g. GitHub). See the Nature Portfolio guidelines for submitting code & software for further information.

## Data

Policy information about availability of data

All manuscripts must include a data availability statement. This statement should provide the following information, where applicable:
- Accession codes, unique identifiers, or web links for publicly available datasets
- A description of any restrictions on data availability
- For clinical datasets or third party data, please ensure that the statement adheres to our policy

Data have been deposited at the NERC Environmental Information Data Centre. Some datasets are currently under embargo until April/May 2025 which can be

lifted if accepted for publication at an earlier date.

Stimuli: "Scanned 3D images and 3D printable images based on combinations of features of Diptera and Hymenoptera collected from the UK in 2021-22"

https://doi.org/10.5285/05169766-7355-4c3c-8ade-091db0583f9d

Wild bird experiments: "Great tit behavioural responses to 3D-printed insect replicas, featuring combinations of traits from wasps and flies, in Madingley Wood, Cambridge, UK, 2021-2023" (under embargo until 1 April 2025)

https://doi.org/10.5285/a1c9b0cc-5585-49c5-a38f-fe05240edccf

Trait Salience: "Chick behavioural responses to 3D-printed insect replicas, featuring combinations of traits from wasps and flies" (under embargo until 1 May 2025)

https://doi.org/10.5285/45391184-603e-4284-bb3c-9c8c6bf856ab

Invertebrate Predators: "Invertebrate behavioural responses to 3D-printed insect replicas, featuring combinations of traits from wasps and flies, in laboratory trials" (under embargo until 1 April 2025)

https://doi.org/10.5285/ee7ba05a-449b-466e-840c-8de1d3f1d4d1

# Research involving human participants, their data, or biological material

Policy information about studies with [underline]human participants or human data[/underline]. See also policy information about [underline]sex, gender (identity/presentation), and sexual orientation[/underline] and [underline]race, ethnicity and racism[/underline].

| Reporting on sex and gender | No human participants/data |
|---|---|
| Reporting on race, ethnicity, or other socially relevant groupings | No human participants/data |
| Population characteristics | No human participants/data |
| Recruitment | No human participants/data |
| Ethics oversight | No human participants/data |

Note that full information on the approval of the study protocol must also be provided in the manuscript.

# Field-specific reporting

Please select the one below that is the best fit for your research. If you are not sure, read the appropriate sections before making your selection.

☐ Life sciences  ☐ Behavioural & social sciences  ☒ Ecological, evolutionary & environmental sciences

For a reference copy of the document with all sections, see [underline]nature.com/documents/nr-reporting-summary-flat.pdf[/underline]

# Ecological, evolutionary & environmental sciences study design

All studies must disclose on these points even when the disclosure is negative.

| Study description | All experiments tested the responses of predators towards 3D-printed stimuli of varying levels of mimetic accuracy, associated with different levels of reward or punishment.

Discrimination Ability: recorded the order in which wild great tits opened dishes bearing 3D stimuli to obtain potential mealworm reward. Three replicate feeding stations with 49 dishes on each, reset and repeated in daily sessions for three weeks. Predictors were 3D stimulus type, presence of mealworm, position of dish (edge vs centre).

Multiple Models experiment: recorded the order in which wild great tits opened dishes bearing 3D stimuli to obtain potential mealworm reward. Six replicate feeding stations divided between two treatments: "one model" or "two model" based on the number of model (unrewarding) stimulus types used. 49 dishes at each feeder, reset daily over the course of six weeks. Predictors were treatment, 3D stimulus type, presence of mealworm, position of dish (edge vs centre).

Validation test: Recorded the order in which wild great tits opened dishes bearing 3D stimuli to obtain potential mealworm reward. Four replicate feeding stations with 24 dishes on each, reset and repeated in daily sessions for five days. Predictors were 3D stimulus type, presence of mealworm, position of dish (edge vs centre).

Trait Salience experiment: recorded the time taken for chicks to approach and peck at dishes bearing 3D stimuli and (for some stimuli) containing a mealworm. 1445 presentations to 30 chicks. Predictors were chick id, day within experimental run (1-4), batch (two different batches of chicks tested on different dates), whether or not it was the first presentation of a trial, and 3D stimulus type.

Invertebrate Predators experiment: recorded behavioural responses of praying mantises (n= 40 presentations, 8 individuals), jumping spiders (n = 57 presentations, 9 individuals) and crab spiders (n = 50 presentations, 50 individuals) towards 3D printed stimuli. Mantises and jumping spiders received all different types of 3D stimulus, whereas crab spiders were randomly assigned to a single stimulus type. Predictors were stimulus type and individual id. |
|---|---|
| Research sample | Discrimination Ability, Multiple Models experiments and Validation test: a population of great tits (Parus major) within Madingley Wood, Cambridge, UK, some PIT tagged (estimated 51-71%). Birds that visited our feeding stations were all adult plumage, but sex and exact age were not recorded. This population was chosen as consisting of wild generalist insectivorous predators expected to show propensity and ability to learn to obtain food from our experimental set up.

Trait Salience: domestic chicks (Gallus gallus domesticus) obtained from a commercial hatchery on the day after hatching, with testing phase of the experiment when they were 17-20 days old. At this age they were not identifiable to sex. They were chosen as |

generalist insectivorous predators that could be housed in a laboratory from hatching and therefore of known prior experience regarding food types. Invertebrate Predators: Praying mantises of three species (Rhombodera kirbyi (n = 5), Polyspilota aeruginosa (n = 1) and Pseudoxyops perpulchra (n = 2)), jumping spiders (Phidippus audax) and crab spiders (Synema globosum). Crab spiders included both male and female, others were not sexed. These subjects were chosen to represent a range of visual types and predation strategies from invertebrates that are known or suspected to prey on Diptera and Hymenoptera.

| | |
|---|---|
| Sampling strategy | Discrimination Ability and Multiple Models experiments, Validation test: sample sizes were constrained mainly by the birds' seasonal behaviour and the number of birds present in the population that chose to visit our feeding stations.<br>Trait Salience and Invertebrate Predators experiments: sample sizes were calculated to give repeats of around n=10 for each stimulus type, minimising the numbers of laboratory animals required. |
| Data collection | Discrimination Ability and Multiple Models experiments, Validation test: video recordings of bird behaviour were taken using a motion-sensitive trail camera and combined with manual records taken of which dishes were open at the end of a given session. Data were recorded by CHT, DB, SB, AC, KC, JRD, SRG, EO, ALP and TR.<br>Trait Salience experiment: behavioural data were noted by the experimenters (CHT and DJGW) at the time of the experiment, and videos were recorded which were later analysed by CHT and DB to determine latency to attack the dishes.<br>Invertebrate Predators: instances of particular behaviours were noted by the experimenters during trials and in some cases verified using video recordings. Data were recorded by TD and RL (jumping spiders), HR (praying mantises), JD, HJJ, JR, JSa (crab spiders). |
| Timing and spatial scale | Discrimination Ability experiment: experiment ran from December 2021 to May 2022, with the testing phase beginning on 5th April 2022. Feeders were spaced within an area approximately 150 m x 150 m.<br>Multiple Models experiment: ran from October 2022 to March 2023, with the testing phase beginning on 13th January 2023.<br>Validation test: ran from October to December 2023, with the testing phase beginning on 18th December. |
| Data exclusions | Discrimination Ability experiment: an additional two feeders were removed from the experiment before the start of the testing phase as they did not receive sufficient great tit visits.<br>Multiple Models experiment: some dishes were opened by mice, which were excluded from the analysis.<br>Validation test: no exclusions<br>Trait Salience experiment: we drew a predetermined cut-off of 13 out of 16 successes during a single trial in the training phase. Chicks that did not reach this cut-off did not proceed to the testing phase. We also excluded testing trials where there was <0.1s in latency to attack towards the fly and wasp stimuli, on the basis that chicks were not showing detectable discrimination among stimulus types.<br>Invertebrate Predators experiment: no exclusions. |
| Reproducibility | No extra tests of reproducibility were carried out beyond the repetition inherent in the experimental design described above. |
| Randomization | Discrimination Ability and Multiple Models experiments, Validation test: within each feeder, a defined set of stimuli were allocated to dishes at random. Treatments for the Multiple Models experiment were assigned to feeders according to spatial grouping, to minimise the numbers of birds that would visit both treatment types.<br>Trait Salience: each trial included presentations of 6 fly and 6 wasp stimuli, as well as 4 probe types that were selected at random from a defined pool (without replacement, to ensure roughly equal sample sizes for each type). Within a trial, order was randomised with the constraint that each half (8 presentations) would contain 3 fly, 3 wasp and 2 probe presentations, to prevent the chicks experiencing long runs of the same stimulus type or reward status.<br>Invertebrate Predators: for mantises and jumping spiders, order of trials was randomised within the testing phase. For crab spiders, individuals were assigned to treatments at random. |
| Blinding | Data transcribed from video recordings were carried out with reference to dish location or trial number and without knowledge of the stimulus type in question (video resolution was not sufficient to reliably identify stimuli visually). |

Did the study involve field work?  ☒ Yes   ☐ No

# Field work, collection and transport

| | |
|---|---|
| Field conditions | Highly variable. During one period of the Multiple Models experiment data collection was paused as snow and below-zero temperatures were causing the dishes to freeze shut. Detailed weather records were not maintained due to the long period of data collection. |
| Location | Madingley Wood, Cambridgeshire, UK (52.217 N, 0.049 E) |
| Access & import/export | Permission for carrying out fieldwork was obtained from University Farm & Rural Estate (University of Cambridge) who manage the woodland |
| Disturbance | Disturbance from researcher visits was minimised as most data recording was done by trail cameras and each feeder was visited only once per day, with visits taking about half an hour per feeder. |

# Reporting for specific materials, systems and methods

We require information from authors about some types of materials, experimental systems and methods used in many studies. Here, indicate whether each material, system or method listed is relevant to your study. If you are not sure if a list item applies to your research, read the appropriate section before selecting a response.

## Materials & experimental systems

| n/a | Involved in the study |
|---|---|
| ☒ | ☐ Antibodies |
| ☒ | ☐ Eukaryotic cell lines |
| ☒ | ☐ Palaeontology and archaeology |
| ☐ | ☒ Animals and other organisms |
| ☒ | ☐ Clinical data |
| ☒ | ☐ Dual use research of concern |
| ☒ | ☐ Plants |

## Methods

| n/a | Involved in the study |
|---|---|
| ☒ | ☐ ChIP-seq |
| ☒ | ☐ Flow cytometry |
| ☒ | ☐ MRI-based neuroimaging |

# Animals and other research organisms

Policy information about studies involving animals; ARRIVE guidelines recommended for reporting animal research, and Sex and Gender in Research

| Laboratory animals | Domestic chicks (Gallus gallus domesticus) were obtained from a commercial hatchery on the day after hatching, with testing phase of the experiment when they were 17-20 days old. Praying mantises of three species (Rhombodera kirbyi (n = 5), Polyspilota aeruginosa (n = 1) and Pseudoxyops perpulchra (n = 2)) and jumping spiders (Phidippus audax) were obtained from commercial sellers at a range of ages from 3rd instar to adult. |
|---|---|
| Wild animals | Wild birds were free to come and go from feeding stations at will, they were not captured at any point during the experiment. Crab spiders (adult, exact age unknown) were captured from the wild by hand in specimen tubes around the Quinta de São Pedro field research station (38.568 N, 9.193 W) and surrounding areas of Sobreda, Portugal, and transported on foot. They were kept for approximately 48 h before being released in the same location. |
| Reporting on sex | Most of our study organisms were not sexable at the time of experiment. Those where sex was identifiable consisted of a mix of sexes, and we have no reason to assume sex biases in the other cases. We did not analyse effects of sex as this was not a variable of interest and would only have been possible in limited sub-sets of the data. |
| Field-collected samples | No laboratory samples collected from the field |
| Ethics oversight | The Trait Salience experiment was approved by Newcastle University AWERB committee (project ID 966). Wild bird experiments (Discrimination Ability and Multiple Models) were approved by AWERB committees at University of Nottingham (project ID 260) and University of Cambridge (ref. NR2022/60). |

Note that full information on the approval of the study protocol must also be provided in the manuscript.

# Plants

| Seed stocks | *Report on the source of all seed stocks or other plant material used. If applicable, state the seed stock centre and catalogue number. If plant specimens were collected from the field, describe the collection location, date and sampling procedures.* |
|---|---|
| Novel plant genotypes | *Describe the methods by which all novel plant genotypes were produced. This includes those generated by transgenic approaches, gene editing, chemical/radiation-based mutagenesis and hybridization. For transgenic lines, describe the transformation method, the number of independent lines analyzed and the generation upon which experiments were performed. For gene-edited lines, describe the editor used, the endogenous sequence targeted for editing, the targeting guide RNA sequence (if applicable) and how the editor was applied.* |
| Authentication | *Describe any authentication procedures for each seed stock used or novel genotype generated. Describe any experiments used to assess the effect of a mutation and, where applicable, how potential secondary effects (e.g. second site T-DNA insertions, mosiacism, off-target gene editing) were examined.* |

