## [Peer Review File · Nature]

Mapping the adaptive landscape of Batesian mimicry using 3D printed stimuli

Corresponding Author: Dr Christopher Taylor

Version 1:

Reviewer comments:

Referee #1

(Remarks to the Author)

This is a fascinating study that provides a much-needed test of multiple dimensions of hoverfly mimicry in the wild using relevant predators. I would argue that the authors could make a bigger deal of their findings on 190-193 that the eye-of-the-beholder hypothesis has less explanatory value in the wild than in laboratory systems using humans or pigeons as proxy predators. The most exciting part of the paper might be that on invertebrate predators, whose ability to discriminate amongst mimetic phenotypes of realistic stimuli has never been so comprehensively explored before (to my knowledge). Also call this out in 82-90 and the summary!

This study will be very helpful in how we think about the evolution of animal communication in diverse, noisy communities, and provides a sweeping view of predator behavior across the adaptive landscape of mimicry. This emphasizes critical differences from previously published work in more idealized systems that has gone on to be considered classic.

38-40: would be helpful to inject some directionality to this statement

60: Kikuchi et al. 2019 eLife make this case for Batesian mimicry, rather than Mullerian as with the Ihalainen paper.

113: maybe "axes" instead of "transects"

212: can you explain better the predictions for these stimuli that are "supernormal" for one model or the other?

Referee #2

(Remarks to the Author)

Review of "Mapping the adaptive landscape for Batesian mimicry using novel 3D-printed stimuli" by Taylor et al.

In this paper the authors use novel 3D-printed reproductions that vary in their similarity to non-mimetic flies and stinging wasps to test several different theories as to how "imperfect mimicry" is maintained by natural selection. Four experiments are described, namely:

- (a) A field experiment evaluating the ability of great tits to discriminate between rewarding mimics (which could be considered as simulated hoverflies) that vary in their distances from rewarding non-mimics (non-mimetic flies) and non-rewarding models (wasps) that they had previously been trained to attack/avoid. Attack order is used as an endpoint.
- (b) A field experiment to test whether rewarding mimics gain the greatest protection from great (and occasional blue) tits by being intermediate in appearance to two non-rewarding models. Attack order is used as an endpoint.
- (c) A lab experiment to elucidate the traits (either colour, pattern, shape, size) used by domestic chicks when differentiating between rewarding and non-rewarding stimuli. All probes were attacked, so attack latency (despite small differences) is used as an endpoint.
- (d) A lab experiment to evaluate the level of protection afforded to non-mimics, mimics and models when presented with three different arthropod species (mantises, jumping spiders, crab spiders). Attack latency and behavioral responses were used as endpoints.

From these experiments, the authors concluded (a) the more wasp-like the stimulus, the less they were prioritized as food by birds (b) intermediate mimics gained no more protection from predation than mimics that differed to the same extent from a

single model (c) chicks tend to use colour and size to discriminate mimics from models and (d) invertebrate predators may also select for mimicry, although they may be less discriminating than birds.

Taken together, this is an impressive series of experiments. The imaginative use of 3D printing has allowed the researchers to characterize the adaptive landscape in a manner that has not previously been investigated. While the headline results (e.g. see Abstract, lines 34-40) that (i) birds can act on subtle differences between phenotypes (seemingly using colour and size), (ii) intermediate mimics gain no special protection from predators and (iii) invertebrate predators can also select for mimicry are not unexpected, the collection of all these findings in one place using the same 3D reproductions makes this a high-quality paper that will be well cited. With four separate experiments there are many moving parts, but we (I have shared this review with a senior postdoc) consider the manuscript well presented and argued. In reviewing the paper, we had several concerns and questions which we hope will be of benefit to the authors if/when revising their paper.

1. One of our biggest concerns is the lack of any detailed consideration of the role of learning. This is despite the fact that the Abstract (line 34) states that "We found that birds have an excellent ability to learn". As we understand it, training in the discrimination experiment took place over 3 weeks and testing took place over 5 weeks. Rather than simply conduct two separate binomial tests for the day 1 vs day 21 of training and qualitatively contrast them (compelling, but not the whole story) why not also see (and test) how the responses vary with training day (through fitting some GLMM model for instance). Just how quickly do they learn? Likewise, data for the first 9 days of the testing phase were excluded (line 935) "while birds adapted to the new set of stimuli". What goes on during learning can still shape selection for mimicry (indeed it is the basis of Müllerian mimicry), so it would help to know more, even if it is retained in the SI. For example, are the novel phenotypes initially prioritized by the birds ("hey those are new, let's try one")? At very least it would help readers to see why 9 days was considered the appropriate cut off point (all we see in Extended Data Figure 1 are days 1-3 vs days 10+). The same reservations apply to the multiple models experiment (where training was twice as long at 6 weeks, and testing was for 10 weeks). Indeed, we are told next to nothing about learning in this experiment except for the lines (223-225) "Once visiting birds were consistently targeting M100 stimuli first, we introduced the intermediate and extrapolated stimuli".

2. It is noteworthy that wild great tits showed no aversion to the wasp-like 3D-printed models compared to the non-mimetic flies at the inception of the experiment (line 137). This indicates that either: 1. these birds are ecologically irrelevant to selection on mimetic phenotypes in hoverflies since they never experience the defenses of wasps and don't know to avoid them or 2. the 3D printed wasps (and/or flies) were readily distinguishable from the real thing. Given that birds are widely recognized as important predators and flies/wasps will be common in Cambridge, then explanation 2 appears the more likely. If this is the case, then while the experiment demonstrated that the birds are capable of learning to discriminate fine details between different stimuli (see below), these results do not translate as directly as one might anticipate to wasps and their mimics under field conditions. Acknowledging that the 3D-phenotypes may potentially be perceived as different from real insects (perhaps because they do not move, or are shinier etc etc) would help to contextualize these investigations.

3. The "level of protection" metric (see for example Figure 2b) needs to be better explained because it has some pathologies that need to be recognized. First, it implicitly assumes that birds approach dishes in the order in which they believe there will be food, which might be the optimal thing to do when other birds are competing for the same resource, but immaterial when a bird has ample time and smorgasbord of choices. Second, within any given trial the data will lack independence, because any prey type with higher scores (such as the consistently unrewarding wasps) will automatically generate lower scores for the other phenotypes. This may be a desirable feature, but the lack of independence might have implications for the underlying tests and how the bootstrapped CIs are interpreted (non-overlapping CIs may not necessarily imply strong evidence of a difference). What do the results look like if one removes the non-mimetic flies and wasps entirely from the sequence which drive the first and last choices respectively: is there still strong evidence of birds prioritizing M75V25 over M25V75 say? Third, it is noted (line 922) that the attack sequence from first to last dish attacked is first converted to a 0-1 proportion scale (0- first dish, 1 - "last or not at all"). These values (which we can call p) were then transformed on a logit scale i.e. $\log(p/(1-p))$. However $\logit(0)$ and $\logit(1)$ are $-\infty$ and $+\infty$, so this transform is inconsistent with a scale that treats 0 as first and 1 as last or not at all. Are the authors using the logit transformed mean p per phenotype per trial, which will rarely be 0 or 1?

4. We appreciate the experimental constraints, but it is worth emphasizing that small differences between phenotypes might be more obvious when the contrasting phenotypes are placed side-by-side on a board compared to sequentially presented as they would be in nature. Indeed, there are some data on this (e.g. Beatty & Franks 2012 Discriminative predation: Simultaneous and sequential encounter experiments, *Current Zoology* 58, 649-657; Kuklová et al. 2023 Does the type of task affect prey discrimination learning in avian predators? *Ethology*, 129, 527-540). Moreover, discrimination might be even more challenging in the real world when the high escape ability of flies means that predators have little time to closely inspect their prey.

5. The invertebrates were: (a) sequentially rather than simultaneously presented with prey, (b) punished rather than not rewarded when attacking a wasp and (c) assessed on the basis of latency to attack rather than attack itself. So, naturally one should be cautious when comparing the outcomes of discrimination tests for invertebrates vs vertebrates. In particular, the argument that invertebrates have (lines 437-438) "less ability [than vertebrates] to discriminate mimics from models" and (lines 442-444) the observation that the experiments have revealed "the incredible discriminatory ability of an insectivorous bird, not matched by important invertebrate predators" clearly needs to be presented with much more nuance. Although a more similar design would have made a fairer comparison, ultimately, these comparisons may prove impossible because, as the authors note, it is all about ecological context.

6. It is stated (line 44) that "Batesian mimics gain protection when predators misidentify them as defended "models"".

However, as is noted elsewhere in the text (see also point 7), the decision to attack a prey item depends on costs and benefits. So, the would-be predator might perceive a prey item to be a fly with high probability, but the costs should it be wrong may be so high that it is not worth the risk. A phrase such as “treat them as if they were defended models” is a longer, but safer, description.

7. The authors ask (line 83) “under ideal conditions, what is the most accurate mimic that a predator can discriminate from its model”. This key question has a distinguished history of consideration going back to RA Fisher (1930, *Genetical Theory of Natural Selection*). Some of this history needs to be recognized because we already know (at least think we know) the answer! Fisher (1930), influenced by psychophysicists, proposed that ANY difference, however small can ultimately be detected on statistical grounds provided enough trials are possible. Duncan & Sheppard (1963, *Continuous and quantal theories of sensory discrimination. Proc. Roy. Soc. B. 158, 343-363*) supported this view experimentally by showing that, on average, humans showed evidence as a group of being able to detect a difference between two weights as small as only 0.125 grams. However, Duncan & Sheppard (see also their 1965 paper, *Sensory discrimination and its role in the evolution of Batesian mimicry. Behaviour 24, 269-282*) also argued (and showed with chicks tasting water) that when binary decisions must be made (such as attack or don't attack) on the basis of perceived differences the response may (or may not) appear quantal depending on the consequences of correct/incorrect responses. As they say “where the consequences are severe, advantages in improving mimicry beyond a certain point will be minimal; whereas if the consequences are mild, the 'quantal effect' will be small and the selective advantage will continue to operate until a perfect resemblance is produced”. Of course, this is an early qualitative invocation of signal detection theory, but it makes clear that given enough time and motivation, one could train birds (and perhaps even invertebrates) to distinguish even the tiniest of differences – there may be no upper threshold beyond perfect mimicry. It is helpful to bear these historical lessons in mind (we appreciate there is some discussion of the role of costs already in the paper) because: (i) the fine discrimination observed in the field experiment was likely only possible given the lengthy training and test periods and the very low cost of mistakes and (ii) it will help avoid questionable statements like (line 191) “improvements in mimetic accuracy would be required to reach the point at which birds are unable to discriminate them from models” because there may be no such point. To re-iterate, its not about telling them apart (as Fisher 1930 argued, a group of predators can exhibit statistical discrimination based on even the tiniest difference) it is about maximizing payoffs given uncertainty.

8. With regard to the multi-model experiment, the hybrid phenotypes e.g. A50V50 and A150V50 are created by interpellation/extrapolation between the two models but where do these phenotypes sit in relation to the positive M100 phenotype they have been trained to find rewarding? Are these hybrids equally distant from the M100s? This is an important question to ask because a simple way to discriminate the rewarding stimulus from the non-rewarding stimulus in the training phase (see Figure 3) is antennae length (and there is evidence that pigeons use this trait too: see Bain et al. 2007. *The key mimetic features of hoverflies through avian eyes. Proc R Soc B. 274, 1949–1954*). Yet the intermediates all have long antennae despite being flies! So, while they are intermediate in appearance between different non-rewarding models, they cannot be thought of as if they had evolved from non-mimetic flies because they were not generated from them. This might help explain why all the wasp-like phenotypes have relatively high levels of protection compared to the M100s. Indeed, despite the stats on lines 239-244, its hard to see evidence of maximal protection around the two models in Figure 3b. In hindsight, would a better design have placed the intermediates somewhere between 3 corners of a triangle (two models and a non-mimic) rather than between 2 points?

9. On line 36 it says “Contrary to prediction” but the prediction comes from Edmunds (ref 10). Indeed ref 6 (Sherratt) states the opposite “When there are two different sympatric model species, then mimics should usually evolve a phenotypic similarity to one or the other model species, but not to both” because the simple signal detection model he outlined suggests that there should be no “special protection” given to intermediates!

10. The 3D models “most closely matched the leg colour of real specimens” (line 631). Note that several of the better hoverfly mimics have black forelegs (because they are waved in front of the head to resemble antennae of hymenoptera), but otherwise yellow legs (see for example Penney et al. 2014. *The relationship between morphological and behavioral mimicry in hoverflies (Diptera: Syrphidae). Am Nat 183*). Having a continuous blend is the way to go by default, but it is worth bearing in mind that some of the higher fidelity mimics are not a continuous blend between the yellow legs of wasps and the black legs of *Mesembrina* (say) but something of a chimera.

11. The exclusion of 12 chicks that were unable to achieve a learning threshold (line 824) is understandable, but consideration might be given to the fact that the predator population is now a subset, namely the better learners.

12. The authors will be aware of many of these papers, but there is now considerable evidence that invertebrates can learn to avoid punishing prey, and that they avoid mimics of these prey while remaining capable of a relatively high degree of discrimination. There are clearly limits on the number of papers that can be cited, but here is a sample from that literature. Gelperin 1968. *Feeding behaviour of the praying mantis: a learned modification. Nature. 219: 399–400*; Berenbaum & Miliczky 1984. *Mantids and milkweed bugs: efficacy of aposematic coloration against invertebrate predators. Am Midl Nat. 111: 64–8*; Taylor et al. 2015 *Flexible color learning in an invertebrate predator: Habronattus jumping spiders can learn to prefer or avoid red during foraging. Behavioral Ecology 27, 520-529*; Bowdish & Bultman 1993 *Visual cues used by mantids in learning aversion to aposematically colored prey. American Midland Naturalist 129, 215-222*; Morris & Reader 2016 *Do crab spiders perceive Batesian mimicry in hoverflies? Behavioral Ecology 27, 920-931*.

(Remarks to the Author)

A. Summary of the key results

This is an amazing contribution to the explanation of Batesian mimicry using a novel idea of imitation of non-existing phenotypes. The authors used 3D prints of artificial prey to create novel mimics (with altered phenotypes) and tested them with several real predators. They used both vertebrate and invertebrate predators. They performed four main experiments each to test a different hypothesis on mimetic accuracy. They conclude that the predator community is the most driving force as postulated in the multiple predator hypothesis.

B. Originality and significance: if not novel, please include reference

The idea to use artificial mimetic phenotypes is completely novel. I particularly like the use of several different predators (which have been rarely used before) because all mimics are occurring in the community of predators. Testing several hypotheses is of major significance.

C. Data & methodology: validity of approach, quality of data, quality of presentation

Methodology used appears to be appropriate, most obtained data of high quality (but see below), data presentation is clear.

D. Appropriate use of statistics and treatment of uncertainties

Most statistical analyses seem appropriate (but see one of my comments below).

E. Conclusions: robustness, validity, reliability

Conclusions are robust.

F. Suggested improvements: experiments, data for possible revision

I have none.

G. References: appropriate credit to previous work?

Mostly, yes but see my major comment below.

H. Clarity and context: lucidity of abstract/summary, appropriateness of abstract, introduction and conclusions

Overall, the manuscript is well prepared, with all details needed provided either in the main text or in supplements.

My major concern is that the Multiple predator hypothesis has been coined and suggested by Pekar et al. (2011) which is not mentioned at all. Instead, it is wrongly attributed to Cuthill & Bennett (1993). Their paper has nothing to do with multiple predator hypothesis as they only argued that pigeons could perceive mimics differently than humans and use their 'the eye of the beholder' hypothesis to explain different ranking of syrphids by pigeons and humans. Yet, neither pigeons nor humans are predators of syrphid flies!

My other concern is related to the results of the trials with spiders. The trial with Phidippus were conducted 30 h after feeding. If the spiders were satiated then this seems a very short time to initiate foraging motivation. This is most likely why the proportion of capturing spiders was so low. Our experience tells us that generalists need about 3 days to become moderately hungry (depending on the ambient temperature, of course). However, even more I am concerned about the trials with Synema. Hunger period of 48 hours at 22 degree Celsius is very short. I am not surprised to read that the capture frequency was so low. In addition, from the video I wonder I doubt that the behaviour of these spiders was foraging (see below).

In the Discussion, I miss information on other signals that mimics and their models are producing (buzzing, smell) which are more important for many arthropod predators than vision. These signals might be more aposematic and informative of prey type than visual signals and alert the predators. Put in another way, experienced predators, such as spiders used in this study, may perceive dangerous wasp only if the visual stimulus is accompanied by sound signals.

Specific comments:

Lines 64-65: This is only one solution but limited because predators and their perception have also evolved.

Lines 66-68: The statement needs to be supported by a citation.

Figures 2, 3: The ordinate is on a logit scale which is difficult to read as it has no bounds. Transform it to probability (using antilogit).

Figure 3. In both experiments wild great tits were used yet the responses were different for the same stimuli. For example, M100 had values below -1 (on logit scale) in the first experiment but about -0.5 in the second experiment. What has happened, why such a big discrepancy? Explain it.

Lines 313-314: Why non-parametric test was used? Clearly, the high number of presentations shows there were repeated use of chicks which should be treated as random effects in time to event analysis.

Lines 348, 418. The Eye of the beholder hypothesis is completely inappropriate here. Replace by a paper on multiple-predator hypothesis.

Lines 400-404: The authors seem to completely ignore other senses of arthropod predators to explain the observed pattern, yet these are often more important than vision. Namely the olfaction and seismic senses are very important. In addition, the movement even though perceived by vision plays a significant role.

Line 754: Only here the reader is informed that the data presented earlier come from both blue and great tit. But on line 216 only great tits are mentioned.

Line 367: I was particularly curious to see how the authors could make a sit-and-wait predator (Synema spiders) to attack a plastic mimic. From the video I could see that the spider is reaching out for a 3D model but I doubt this is a grabbing

behaviour resembling an attack. It rather looked like a behaviour to escape.

Version 2:

Reviewer comments:

Referee #1

(Remarks to the Author)

The authors have addressed my comments to my satisfaction. I have no further changes to recommend.

Referee #2

(Remarks to the Author)

I have re-reviewed the paper "Mapping the adaptive landscape for Batesian mimicry using novel 3D-printed stimuli" by Taylor et al., along with a postdoc in my lab (in confidence). We continue to think that this is an impressive series of experiments, all linked by the imaginative use of 3D printing.

Our original concerns were: (i) lack of a more detailed analysis of the role of learning, (ii) a comment on the surprising lack of aversion to wasp-like stimuli at the start of the experiment, (iii) some pathologies of the metric used to quantify levels of protection, (iv) the potential difference between sequential and simultaneous presentation, (v) differences in the design of the vertebrate vs invertebrate experiments make comparisons challenging, (vi) the difference between beliefs and actions, (vii) the lack of a statistical limit on phenotypic differentiability, (viii) the role of the distance of the intermediate phenotypes from the rewarding stimulus, along with four other more minor comments.

Having carefully reviewed the revised paper, we are pleased to say that we think the authors have addressed most of our concerns very well. The edits include (a) extended data figures showing changes in preference over time, (b) details of a new validation experiment (along with extended data) involving dead specimens of flies and wasps, (c) a robustness test temporarily removing the X100 phenotypes from the analysis, (d) a justification for adding a small constant to the odds numerator and denominator, and (e) a number of small but important clarificatory edits. The authors have clearly taken every comment (not just ours, but also the other reviewers) seriously. The new validation experiment (line 195 et seq) asked whether birds subsequently generalize their learned avoidance of wasp 3D models to (dead) wasps - they do. This is not quite the same as asking whether the birds would avoid real wasps without training given their prior natural experiences, but it does at least indicate that after training the two types of wasp (3D and real) are treated similarly. Collectively, these changes have made a more nuanced and transparent paper. This is an exciting study, and it is great to see all the material in one paper rather than split over multiple papers where key messages can get lost.

On re-reading the paper, we had a few additional minor concerns that we didn't catch first time around.

i) From the extended data table 2 it appears (if the base formula has edge as the only predictor) that the authors compared the AIC of models that included interactions (such as m:p) but lacked the composite main effects (notably p). This violates the principle of marginality and in so doing generates incorrect estimates of the interaction coefficients (since the interaction term now includes the main effects it is composed of). The lack of hierarchical considerations when fitting models may or may not matter when it comes to model comparison (it may simply be a re-parameterized model), but it is sufficiently important to warrant checking (see Grafen & Hails 2002 Modern Statistics, p. 192). Indeed, some software prevents analysts from fitting models with interactions without all their constituent main effects.

ii) On the subject of interactions, why 2 df with the intermediate:distance and yet 1 df with the treatment:distance?

iii) Given the potential role of "edge" (outer perimeter), we assume that treatments were randomised with respect to edge?

iv) A rose by any other name smells just as sweet, but on second reading we were also temporarily confused by the term "transect", which is suggestive of stimuli being laid out in a line.

Referee #3

(Remarks to the Author)

The authors responded appropriately to all my former comments, improved or clarified most of the text. After reading the manuscript again I have few more concerns that mainly concern the statistical analyses.

The multiple models hypothesis seems to be misinterpreted here. You predict greater protection for an intermediate mimic (l. 231). This is not in agreement with Edmunds (1991) who states that the poor (=intermediate) mimic gains higher protection NOT on the individual BUT population level. In other words, an individual of a poor mimic population will have lower (NOT HIGHER as you predict) protection from predators compared to a good mimetic individual because it resembles more models so the population of the poor mimic can occur on a larger spatial and/or temporal scale. So the whole population (NOT an individual) of a poor mimic will have higher fitness than the population of a good mimic. To test this hypothesis per se you would need to compare POPULATIONS of poor and good mimics in temporal and spatial scale(s). This was not done in your experiment as you only compared resemblance to two models. Yet, even your experiment has some merit for the Multiple models hypothesis. The results of your experiment, in contrast to what you conclude, support the Multiple models hypothesis, in my opinion, because the protection of intermediate (poor) mimics was not significantly different from

protection of good mimic. So you would need to reformulate the predictions and conclusions of this experiment.

L. 527: Or they use different stimuli – acoustic, seismic or chemical.

I looked at the statistical analyses in more detail and I have the following comments. Overall, the analyses appear to be correct. I use the word appear because not all explanatory variables are defined (namely, reward, edge, stimulus, treatment) thus it was not possible to check whether the linear predictor of the fitted models make sense. I believe they do, though.

Further, I was surprised why authors estimated confidence intervals using bootstrapping. If the design of the experiments was completely random without (dependencies) pseudoreplications then the use of bootstrap would be fine. But the design was more complicated (nested), then the bootstrap estimates might not be better than estimates from the linear models. It depends how bootstrapping was done. Did authors did not believe the estimates (mean values and their standard errors) of the model? They do not describe nor explain why bootstrapping was used.

The protection values were logit transformed which is OK. However, then it is not clear which family has been used in GLM analysis. Such values are not from binomial distribution so the use of binomial family is not suitable and beta should be used. Beta is, however, not implemented in many packages and from the text of methods it is not clear which package for GLM and GLMM authors used (l. 1057).

The authors fitted two models using NLS regression which is not trivial to use. Indeed, for some data the algorithm did not converge. However, I miss arguments why these two models were used. Typically, this is done because some (or all) model parameters have biological meaning. But this was not the case which makes me wonder why they did not use linear models for curvilinear relationships, instead, which are easier to fit. The sigmoid 3-parameter model (l. 1082) could be replaced by 2-parameter sigmoid (logistic) model implemented within GLM and binomial errors. Similarly, the 3-parameter asymptotic model could be replaced by 2-parameter (Michaelis-Menten) model fit with GLM with Gamma errors.

L. 1119: How did you deal with feeders in the GLM models? On l. 1099 you state that feeder was fitted as fixed effect due to low number of feeders (I agree) but then on l. 1119 you state it was fitted as random effect, yet the number of feeders was the same.

Version 3:

Reviewer comments:

Referee #1

(Remarks to the Author)

I had no comments on this round of revision, and am happy for the ms to be published.

Referee #2

(Remarks to the Author)

I have re-reviewed "Mapping the adaptive landscape.." by Taylor et al., focusing in particular on my follow up minor comments concerning marginality and separating main effects from interactions. I'm happy to say I think the authors have given a very reasoned response and that these points have been clarified in the text. As stated in the original review, this is an impressive series of experiments, covering a lot of ground and all linked by the imaginative use of 3D printing.

Referee #3

(Remarks to the Author)

The authors have responded to all my comments and changed the text of the manuscript satisfactorily. I have no other comments.

We are delighted that the referees appreciate the novelty and significance of our work, and thank them for their detailed and insightful comments. We have included some extra data and analysis, as well as changes to the text, which we believe address the comments and improve the quality of the manuscript. In particular, we have added data from a supplementary experiment to support our statements about the realism of our stimuli (see R2 point 2 for details). Below, we respond to each point in turn. Line references are based on the “track changes” version of the document with “all markup” selected.

Referee #1 (Remarks to the Author):

This is a fascinating study that provides a much-needed test of multiple dimensions of hoverfly mimicry in the wild using relevant predators. I would argue that the authors could make a bigger deal of their findings on 190-193 that the eye-of-the-beholder hypothesis has less explanatory value in the wild than in laboratory systems using humans or pigeons as proxy predators. The most exciting part of the paper might be that on invertebrate predators, whose ability to discriminate amongst mimetic phenotypes of realistic stimuli has never been so comprehensively explored before (to my knowledge). Also call this out in 82-90 and the summary!

Thank you for the positive evaluation of the manuscript, and the suggestions to emphasise the invertebrate results. We have now called out this point in the discussion (line 503-504), and we also now give a more detailed discussion of how our work fits in with the existing literature on invertebrate responses to mimicry (line 406-421) – see also our response to R2, point 12 below.

This study will be very helpful in how we think about the evolution of animal communication in diverse, noisy communities, and provides a sweeping view of predator behavior across the adaptive landscape of mimicry. This emphasizes critical differences from previously published work in more idealized systems that has gone on to be considered classic.

Thank you for the positive evaluation.

38-40: would be helpful to inject some directionality to this statement

We have clarified the direction of variation (line 41-42).

60: Kikuchi et al. 2019 eLife make this case for Batesian mimicry, rather than Mullerian as with the Ihalainen paper.

This is a useful and relevant reference and we have added it (line 64).

113: maybe “axes” instead of “transects”

We did deliberate over the best choice of term here, and agree with the referee that “axis” is a strong contender. Indeed, in our original submission, we used the term as part of our definition of what we mean by a transect (new line numbers 710-715). Our

reason for choosing “transect” was that we felt it better captures the idea that we are sampling from within a complex multidimensional phenotypic space, in a way that is defined by its endpoints rather than a particular variable or trait. We were also concerned to avoid the implication (even if not absolutely stipulated by the term) that our three “axes” of mimetic accuracy (that use *Mesembrina*, *Syrphus* and *Chrysotoxum* as starting points) were orthogonal to each other, which they are not.

Because of this, we would marginally prefer to keep the term “transect” here, but do not have a hard position on it, so if the editor would prefer, we would be happy to replace with “axis”.

212: can you explain better the predictions for these stimuli that are “supernormal” for one model or the other?

We have added an example of comparison between an intermediate and an extrapolated mimic to clarify our prediction (line 243-247).

Referee #2 (Remarks to the Author):

Review of “Mapping the adaptive landscape for Batesian mimicry using novel 3D-printed stimuli” by Taylor et al.

In this paper the authors use novel 3D-printed reproductions that vary in their similarity to non-mimetic flies and stinging wasps to test several different theories as to how “imperfect mimicry” is maintained by natural selection. Four experiments are described, namely:

(a) A field experiment evaluating the ability of great tits to discriminate between rewarding mimics (which could be considered as simulated hoverflies) that vary in their distances from rewarding non-mimics (non-mimetic flies) and non-rewarding models (wasps) that they had previously been trained to attack/avoid. Attack order is used as an endpoint.

(b) A field experiment to test whether rewarding mimics gain the greatest protection from great (and occasional blue) tits by being intermediate in appearance to two non-rewarding models. Attack order is used as an endpoint.

(c) A lab experiment to elucidate the traits (either colour, pattern, shape, size) used by domestic chicks when differentiating between rewarding and non-rewarding stimuli. All probes were attacked, so attack latency (despite small differences) is used as an endpoint.

(d) A lab experiment to evaluate the level of protection afforded to non-mimics, mimics and models when presented with three different arthropod species (mantises, jumping spiders, crab spiders). Attack latency and behavioral responses were used as endpoints.

From these experiments, the authors concluded (a) the more wasp-like the stimulus,

the less they were prioritized as food by birds (b) intermediate mimics gained no more protection from predation than mimics that differed to the same extent from a single model (c) chicks tend to use colour and size to discriminate mimics from models and (d) invertebrate predators may also select for mimicry, although they may be less discriminating than birds.

Taken together, this is an impressive series of experiments. The imaginative use of 3D printing has allowed the researchers to characterize the adaptive landscape in a manner that has not previously been investigated. While the headline results (e.g. see Abstract, lines 34-40) that (i) birds can act on subtle differences between phenotypes (seemingly using colour and size), (ii) intermediate mimics gain no special protection from predators and (iii) invertebrate predators can also select for mimicry are not unexpected, the collection of all these findings in one place using the same 3D reproductions makes this a high-quality paper that will be well cited. With four separate experiments there are many moving parts, but we (I have shared this review with a senior postdoc) consider the manuscript well presented and argued. In reviewing the paper, we had several concerns and questions which we hope will be of benefit to the authors if/when revising their paper.

Thank you for the positive evaluation of the manuscript.

1. One of our biggest concerns is the lack of any detailed consideration of the role of learning. This is despite the fact that the Abstract (line 34) states that “We found that birds have an excellent ability to learn”. As we understand it, training in the discrimination experiment took place over 3 weeks and testing took place over 5 weeks. Rather than simply conduct two separate binomial tests for the day 1 vs day 21 of training and qualitatively contrast them (compelling, but not the whole story) why not also see (and test) how the responses vary with training day (through fitting some GLMM model for instance). Just how quickly do they learn? Likewise, data for the first 9 days of the testing phase were excluded (line 935) “while birds adapted to the new set of stimuli”. What goes on during learning can still shape selection for mimicry (indeed it is the basis of Müllerian mimicry), so it would help to know more, even if it is retained in the SI. For example, are the novel phenotypes initially prioritized by the birds (“hey those are new, let’s try one”)? At very least it would help readers to see why 9 days was considered the appropriate cut off point (all we see in Extended Data Figure 1 are days 1-3 vs days 10+). The same reservations apply to the multiple models experiment (where training was twice as long at 6 weeks, and testing was for 10 weeks). Indeed, we are told next to nothing about learning in this experiment except for the lines (223-225) “Once visiting birds were consistently targeting M100 stimuli first, we introduced the intermediate and extrapolated stimuli”.

We agree that sharing more data on the learning processes from the great tit experiments would be valuable. We have added this information to the Extended Data showing changes in preference over time in both phases of the Discrimination Ability (ED Figure 1) and Multiple Models (ED Figure 2) experiments. For the Discrimination Ability section, we originally included textual descriptions of changes with time (new line numbers training phase 145-152; testing phase 155-160) – in particular, novel phenotypes were not prioritised: “targeting the least accurate stimuli first and moving

on to explore the more accurate mimics when no other options were available”. We have now added Extended Data Figure 1 so that the reader can see the learning curves for themselves, and we describe the curve-fitting process in the methods (line 1079-1087). Note that we took an illustrative, rather than hypothesis-testing, approach to curve fitting as there are many variants possible based on different formulae or subdivisions of the data, and we did not have a single a priori prediction for the shape of the learning curve.

For the Multiple Models experiment, our original reporting of the learning process was brief, and we have now expanded it to include a figure illustrating changes over time for both phases (Extended Data Figure 2) and a description of the changes with time during the testing phase (lines 274-276). Revisiting these learning data highlighted to us a feature that we had previously glossed over, which is that, in contrast to the Discrimination Ability experiment, levels of discrimination declined during the first 10 days of the testing phase, with interesting implications that we have now discussed in detail in the main text (lines 282-297).

Regarding the question about our choice of 10 days as a cut-off for the learning period, for the Discrimination Ability experiment, we have added clarification that this was based on fitting asymptotic curves to the learning data and taking the point at which all curves were within 10% of their asymptote (line 1090). For the Multiple Models experiment, we have noticed an inconsistency which is that we did not exclude the learning period as we had done for the Discrimination Ability experiment, which we have now rectified. The same process was not possible for the Multiple Models experiment as fitting of asymptotic curves failed to converge, but from visual assessment of the data, the same cut-off appears to be appropriate (explanation added in lines 1104-1116). The implementation of this cutoff alters some of the summary statistics and measures of fit slightly (lines 278-281, Figure 3, Extended Data Table 3) but the ranking of the models remains the same and our conclusions are unaffected.

2. It is noteworthy that wild great tits showed no aversion to the wasp-like 3D-printed models compared to the non-mimetic flies at the inception of the experiment (line 137). This indicates that either: 1. these birds are ecologically irrelevant to selection on mimetic phenotypes in hoverflies since they never experience the defenses of wasps and don't know to avoid them or 2. the 3D printed wasps (and/or flies) were readily distinguishable from the real thing. Given that birds are widely recognized as important predators and flies/wasps will be common in Cambridge, then explanation 2 appears the more likely. If this is the case, then while the experiment demonstrated that the birds are capable of learning to discriminate fine details between different stimuli (see below), these results do not translate as directly as one might anticipate to wasps and their mimics under field conditions. Acknowledging that the 3D-phenotypes may potentially be perceived as different from real insects (perhaps because they do not move, or are shinier etc etc) would help to contextualize these investigations.

We agree that there are several explanations for the lack of aversion to wasp stimuli at the start of the experiment, one of which is that the stimuli were readily distinguished from real wasps. To address this issue, we now present new experimental data comparing responses to printed stimuli and real specimens (lines 195-204,

methods lines 862-871, Extended Data Figure 4). Our stimuli will never be 100% identical to the real insects (now acknowledged in lines 491-493), and although our findings indicate that birds are unlikely to perceive the 3D models and insects as identical in all cases, they clearly generalise from one to the other, indicating that they share a strong visual similarity. We have added a section to discuss the value and limitations of our stimuli, emphasising that they represent a major improvement on models used in mimicry experiments to date (lines 491-502).

Another possible reason for the lack of initial avoidance of the wasp stimuli is that the experiment was conducted during winter and early spring, when wasps are very unlikely to be encountered (now noted in line 148-149). It is plausible that learned avoidance of wasps wanes rapidly and needs to be re-learned each year.

3. The “level of protection” metric (see for example Figure 2b) needs to be better explained because it has some pathologies that need to be recognized. First, it implicitly assumes that birds approach dishes in the order in which they believe there will be food, which might be the optimal thing to do when other birds are competing for the same resource, but immaterial when a bird has ample time and smorgasbord of choices. Second, within any given trail the data will lack independence, because any prey type with higher scores (such as the consistently unrewarding wasps) will automatically generate lower scores for the other phenotypes. This may be a desirable feature, but the lack of independence might have implications for the underlying tests and how the bootstrapped CIs are interpreted (non-overlapping CIs may not necessarily imply strong evidence of a difference). What do the results look like if one removes the non-mimetic flies and wasps entirely from the sequence which drive the first and last choices respectively: is there still strong evidence of birds prioritizing M75V25 over M25V75 say? Third, it is noted (line 922) that the attack sequence from first to last dish attacked is first converted to a 0-1 proportion scale (0- first dish, 1 – “last or not at all”). These values (which we can call p) were then transformed on a logit scale i.e. $\log(p/(1-p))$. However $\text{logit}(0)$ and $\text{logit}(1)$ are $-\infty$ and $+\infty$, so this transform is inconsistent with a scale that treats 0 as first and 1 as last or not at all. Are the authors using the logit transformed mean p per phenotype per trial, which will rarely be 0 or 1?

The amount of food available at a feeder is fixed for a given session, and therefore it is true that an individual bird with unlimited time could get the same total payoff for opening the dishes in any order. However, for several reasons birds are likely to be motivated to maximise their payoff per unit of time, as is often assumed in optimal foraging models (Pyke, Pulliam et al. 1977): the risk of losing food to competitors, increased risk of predation while foraging, and opportunity cost of foraging elsewhere. Thus, even when presented with simultaneous choices, birds are regularly shown to prioritise those with the highest perceived probability of payoff (Davies 1977, Erichsen, Krebs et al. 1980, Lindström, Alatalo et al. 2004). Moreover, the birds do open the dishes in a non-random order, with initial lack of preference ruling out pre-existing biases, so it is difficult to imagine what else this order might represent other than “the order in which they believe there will be food”. We have now added explicit description of these assumptions (line 140-142).

We have carried out the suggested robustness test of removing all M100, C100, S100 and V100 phenotypes from the dataset and calculating new rank orders for dish opening. Analysis of this filtered dataset yielded very similar results to those presented in the manuscript, with post-hoc tests confirming significant differences between [MCS]25V75 and the 50-50 phenotypes. To keep the messaging in the manuscript clear, especially given the various ways in which filtering could be applied to the analysis, we have not added this additional analysis to the manuscript, but we provide it as an appendix to this response to the referees for the sake of transparency.

The logit transform was carried out on individual dishes rather than mean values, but included a small adjustment to account for 0 and 1 values, as suggested by Warton and Hui (2011). We have now added clarification that the formula used was $\log((p + 0.01)/(1-p + 0.01))$ – line 1068-1070.

4. We appreciate the experimental constraints, but it is worth emphasizing that small differences between phenotypes might be more obvious when the contrasting phenotypes are placed side-by-side on a board compared to sequentially presented as they would be in nature. Indeed, there are some data on this (e.g. Beatty & Franks 2012 Discriminative predation: Simultaneous and sequential encounter experiments, Current Zoology 58, 649-657; Kuklová et al. 2023 Does the type of task affect prey discrimination learning in avian predators? Ethology, 129, 527-540). Moreover, discrimination might be even more challenging in the real world when the high escape ability of flies means that predators have little time to closely inspect their prey.

We have edited the discussion of the Discrimination experiment section to acknowledge these factors (lines 217-221).

5. The invertebrates were: (a) sequentially rather than simultaneously presented with prey, (b) punished rather than not rewarded when attacking a wasp and (c) assessed on the basis of latency to attack rather than attack itself. So, naturally one should be cautious when comparing the outcomes of discrimination tests for invertebrates vs vertebrates. In particular, the argument that invertebrates have (lines 437-438) “less ability [than vertebrates] to discriminate mimics from models” and (lines 442-444) the observation that the experiments have revealed “the incredible discriminatory ability of an insectivorous bird, not matched by important invertebrate predators” clearly needs to be presented with much more nuance. Although a more similar design would have made a fairer comparison, ultimately, these comparisons may prove impossible because, as the authors note, it is all about ecological context.

We agree with the referee that there are differences in design of the invertebrate and bird experiments which might provide partial explanations for the differences in outcomes, and our original submission discussed possible effects of positive vs negative conditioning (new line numbers 463-466). This is one reason why we already highlighted differences within the invertebrates (new line numbers line 459-460, 466-469), which are far more similar in experimental design. We have adjusted our wording in the conclusion (line 532-534) to avoid overstating the difference between birds and invertebrates. We also agree with the point that it will never be possible to completely remove such factors in an experimental setting. Despite these limitations, we believe

that comparison among predators is important, and we agree with Referee 1 that “[invertebrate predators’] ability to discriminate amongst mimetic phenotypes of realistic stimuli has never been so comprehensively explored before”.

6. It is stated (line 44) that “Batesian mimics gain protection when predators misidentify them as defended “models””. However, as is noted elsewhere in the text (see also point 7), the decision to attack a prey item depends on costs and benefits. So, the would-be predator might perceive a prey item to be a fly with high probability, but the costs should it be wrong may be so high that it is not worth the risk. A phrase such as “treat them as if they were defended models” is a longer, but safer, description.

We have made the suggested change to the wording (line 48).

7. The authors ask (line 83) “under ideal conditions, what is the most accurate mimic that a predator can discriminate from its model”. This key question has a distinguished history of consideration going back to RA Fisher (1930, *Genetical Theory of Natural Selection*). Some of this history needs to be recognized because we already know (at least think we know) the answer! Fisher (1930), influenced by psychophysicists, proposed that ANY difference, however small can ultimately be detected on statistical grounds provided enough trials are possible. Duncan & Sheppard (1963, *Continuous and quantal theories of sensory discrimination. Proc. Roy. Soc. B. 158, 343-363*) supported this view experimentally by showing that, on average, humans showed evidence as a group of being able to detect a difference between two weights as small as only 0.125 grams. However, Duncan & Sheppard (see also their 1965 paper, *Sensory discrimination and its role in the evolution of Batesian mimicry. Behaviour 24, 269-282*) also argued (and showed with chicks tasting water) that when binary decisions must be made (such as attack or don’t attack) on the basis of perceived differences the response may (or may not) appear quantal depending on the consequences of correct/incorrect responses. As they say “where the consequences are severe, advantages in improving mimicry beyond a certain point will be minimal; whereas if the consequences are mild, the ‘quantal effect’ will be small and the selective advantage will continue to operate until a perfect resemblance is produced”. Of course, this is an early qualitative invocation of signal detection theory, but it makes clear that given enough time and motivation, one could train birds (and perhaps even invertebrates) to distinguish even the tiniest of differences – there may be no upper threshold beyond perfect mimicry. It is helpful to bear these historical lessons in mind (we appreciate there is some discussion of the role of costs already in the paper) because: (i) the fine discrimination observed in the field experiment was likely only possible given the lengthy training and test periods and the very low cost of mistakes and (ii) it will help avoid questionable statements like (line 191) “improvements in mimetic accuracy would be required to reach the point at which birds are unable to discriminate them from models” because there may be no such point. To re-iterate, its not about telling them apart (as Fisher 1930 argued, a group of predators can exhibit statistical discrimination based on even the tiniest difference) it is about maximizing payoffs given uncertainty.

We agree that the maximum level of discrimination reached will depend on the levels of motivation. We have adjusted our wording to avoid reference to an upper

ceiling of discrimination ability (lines 87-89 and 211-214) and added some further discussion of the relative importance of sensory ability versus motivation (lines 209-221) including the useful historical references highlighted by the referees (line 210).

8. *With regard to the multi-model experiment, the hybrid phenotypes e.g. A50V50 and A150V50 are created by interpellation/extrapolation between the two models but where do these phenotypes sit in relation to the positive M100 phenotype they have been trained to find rewarding? Are these hybrids equally distant from the M100s? This is an important question to ask because a simple way to discriminate the rewarding stimulus from the non-rewarding stimulus in the training phase (see Figure 3) is antennae length (and there is evidence that pigeons use this trait too: see Bain et al. 2007. The key mimetic features of hoverflies through avian eyes. Proc R Soc B. 274, 1949–1954). Yet the intermediates all have long antennae despite being flies! So, while they are intermediate in appearance between different non-rewarding models, they cannot be thought of as if they had evolved from non-mimetic flies because they were not generated from them. This might help explain why all the wasp-like phenotypes have relatively high levels of protection compared to the M100s. Indeed, despite the stats on lines 239-244, its hard to see evidence of maximal protection around the two models in Figure 3b. In hindsight, would a better design have placed the intermediates somewhere between 3 corners of a triangle (two models and a non-mimic) rather than between 2 points?*

Our primary aim with this experiment was to test whether an intermediate mimic (i.e. one which is an equal phenotypic distance (let this be “d”) from each of two different models) gains any additional protection compared to one where similarity to one of the models is much lower (=d) than it is to the other (>>d). This contrast is maximised by using an axis of variation that runs through the two models and extends beyond in both directions, as we have used here. We recognise the referee’s point that our mimics here do not represent realistic hoverfly-like phenotypes; instead, they are idealised mimics generated specifically to test the extreme scenario where a mimic is a perfect intermediate. We agree this design may result in some aspects of the model appearance (such as antenna length) being relatively conserved across the mimics, and have now added some discussion on the possible implications for the shape of the adaptive landscape in the discussion (line 291-297).

Other designs would certainly have been possible, and were considered in our design process, such as using transects running from M100 to A100 and from M100 to V100 as suggested by the referee. This would have needed to be in addition to the transect A100 to V100 that we used, or else no “jack of all trades” mimics would have been generated. While an interesting approach, this design would have required a much higher number of stimulus types, limiting experimental power.

Regarding the referees’ comment that “its hard to see evidence of maximal protection around the two models in Figure 3b”, we have added an insert to the figure which groups stimuli according to their distance to the nearest model, to help show the relationship between mimetic accuracy and protection more clearly (Figure 3).

9. *On line 36 it says “Contrary to prediction” but the prediction comes from Edmunds*

(ref 10). Indeed ref 6 (Sherratt) states the opposite “When there are two different sympatric model species, then mimics should usually evolve a phenotypic similarity to one or the other model species, but not to both” because the simple signal detection model he outlined suggests that there should be no “special protection” given to intermediates!

Our reading of Sherratt (2002) is not so absolute. From the main text: “If the models are sufficiently dissimilar from one another ... then any mimic will be attacked least frequently on encounter if it resembled one model, or the other, but not both ... if the two models resemble one another to an extent ... then there will be a range of intermediate phenotypes at which the mimics will be attacked with equally low frequency.” This doesn’t predict an advantage to intermediate mimics over perfect mimics, but equally it does (along with equation 8 from that paper) seem to allow for the possibility that intermediate mimics receive some extra protection compared to that they would receive from a single model.

To avoid any doubt, we have removed the statement “contrary to prediction” from the abstract (line 39), and have corrected the reference within the Multiple Models section (line 227).

10. The 3D models “most closely matched the leg colour of real specimens” (line 631). Note that several of the better hoverfly mimics have black forelegs (because they are waved in front of the head to resemble antennae of hymenoptera), but otherwise yellow legs (see for example Penney et al. 2014. The relationship between morphological and behavioral mimicry in hoverflies (Diptera: Syrphidae). Am Nat 183). Having a continuous blend is the way to go by default, but it is worth bearing in mind that some of the higher fidelity mimics are not a continuous blend between the yellow legs of wasps and the black legs of Mesembrina (say) but something of a chimera.

We have tweaked the wording to clarify that the “most closely matched” refers to the majority leg colour of the specimens (line 745-746). We have also discussed the realism of the stimuli in general terms (line 491-502).

It is true that increasing the resolution of the representation of the legs by splitting them into sections, or applying the distance transform process to each one separately, might have further increased the fidelity of our stimuli. This is something we will bear in mind for future research. We had to draw a line regarding how detailed to attempt our representations, determined by a) limitations of the printing technology, b) the degree of manual manipulation required of the digital files, and c) wanting to limit the number of special cases within the implementation, to limit its subjectivity and maximise the broadness of applicability.

11. The exclusion of 12 chicks that were unable to achieve a learning threshold (line 824) is understandable, but consideration might be given to the fact that the predator population is now a subset, namely the better learners.

We have added acknowledgement of this point (line 962-965). There are many possible reasons for a lack of learning in some individuals, including factors such as

stress or motivation which may be particular to our experiment. The result of predation from a suite of predators including some selective and some not, whether varying within or among species, is that selective pressure will still be exerted on those traits which influence the behaviour of at least some predators, so we do not see a reason to think that our key conclusions would be altered in light of this variation.

12. The authors will be aware of many of these papers, but there is now considerable evidence that invertebrates can learn to avoid punishing prey, and that they avoid mimics of these prey while remaining capable of a relatively high degree of discrimination. There are clearly limits on the number of papers that can be cited, but here is a sample from that literature. Gelperin 1968. Feeding behaviour of the praying mantis: a learned modification. Nature. 219: 399–400; Berenbaum & Miliczky 1984. Mantids and milkweed bugs: efficacy of aposematic coloration against invertebrate predators. Am Midl Nat. 111: 64–8; Taylor et al. 2015 Flexible color learning in an invertebrate predator: Habronattus jumping spiders can learn to prefer or avoid red during foraging. Behavioral Ecology 27, 520-529; Bowdish & Bultman 1993 Visual cues used by mantids in learning aversion to aposematically colored prey. American Midland Naturalist 129, 215-222; Morris & Reader 2016 Do crab spiders perceive Batesian mimicry in hoverflies? Behavioral Ecology 27, 920-931.

We have now expanded on our discussion of current knowledge of response of invertebrates towards aposematic and mimetic prey and added further references (lines 412-419).

Referee #3 (Remarks to the Author):

A. Summary of the key results

This is an amazing contribution to the explanation of Batesian mimicry using a novel idea of imitation of non-existing phenotypes. The authors used 3D prints of artificial prey to create novel mimics (with altered phenotypes) and tested them with several real predators. They used both vertebrate and invertebrate predators. They performed four main experiments each to test a different hypothesis on mimetic accuracy. They conclude that the predator community is the most driving force as postulated in the multiple predator hypothesis.

B. Originality and significance: if not novel, please include reference

The idea to use artificial mimetic phenotypes is completely novel. I particularly like the use of several different predators (which have been rarely used before) because all mimics are occurring in the community of predators. Testing several hypotheses is of major significance.

C. Data & methodology: validity of approach, quality of data, quality of presentation

Methodology used appears to be appropriate, most obtained data of high quality (but see below), data presentation is clear.

D. Appropriate use of statistics and treatment of uncertainties

Most statistical analyses seem appropriate (but see one of my comments below).

E. Conclusions: robustness, validity, reliability

Conclusions are robust.

F. Suggested improvements: experiments, data for possible revision

I have none.

G. References: appropriate credit to previous work?

Mostly, yes but see my major comment below.

H. Clarity and context: lucidity of abstract/summary, appropriateness of abstract, introduction and conclusions

Overall, the manuscript is well prepared, with all details needed provided either in the main text or in supplements.

We thank the referee for the positive evaluation of the manuscript's originality and significance.

1. My major concern is that the Multiple predator hypothesis has been coined and suggested by Pekar et al. (2011) which is not mentioned at all. Instead, it is wrongly attributed to Cuthill & Bennett (1993). Their paper has nothing to do with multiple predator hypothesis as they only argued that pigeons could perceive mimics differently than humans and use their 'the eye of the beholder' hypothesis to explain different ranking of syrphids by pigeons and humans. Yet, neither pigeons nor humans are predators of syrphid flies!

We thank the referee for highlighting this highly relevant paper. We have added reference to this paper and discussion of its importance in lines 410-414 and 512-515. However, we do see some conceptual differences between our hypothesis test and that proposed by Pekar et al. Unlike what is proposed in their Multiple Models hypothesis, we have not tested the effects of a predator that favours attacking models, nor have we tested the combined effects of multiple predators simultaneously attacking the same mimetic prey, and our conclusions do not depend on such scenarios; rather, we simply observe that depending on the predator species, different degrees of mimetic accuracy may be required in order to gain protection. That is where our experiments relate to the Eye of the Beholder hypothesis: we are testing whether a mimic that is imperfect in the eyes of one predator/beholder might be sufficiently deceiving in the eyes of another to mean that imperfect mimicry could persist in a community in which the latter predator is the most important agent of selection on mimetic accuracy.

We have changed the heading of this section from "Multiple Predators" to "Invertebrate Predators" (line 393) to better reflect the key feature of that section and to avoid confusing this with the related but distinct "Multiple Predators hypothesis" as discussed above. We also appreciate the referee's point that the Cuthill and Bennett (1993) paper is quite focused in its scope on differences in the visual abilities of pigeons

and humans. We have made some edits to clarify our use of the phrase “Eye of the Beholder” and better reflect that it has arisen from more than just that one source (line 407-410).

2. My other concern is related to the results of the trials with spiders. The trial with Phidippus were conducted 30 h after feeding. If the spiders were satiated then this seems a very short time to initiate foraging motivation. This is most likely why the proportion of capturing spiders was so low. Our experience tells us that generalists need about 3 days to become moderately hungry (depending on the ambient temperature, of course). However, even more I am concerned about the trials with Synema. Hunger period of 48 hours at 22 degree Celsius is very short. I am not surprised to read that the capture frequency was so low. In addition, from the video I wonder I doubt that the behaviour of these spiders was foraging (see below).

We agree with the referee that motivation to hunt is dependent on a range of factors, including time since the last meal (with size of that meal a relevant factor) and temperature; in our experience, spider sex and developmental stage are very important too. Our jumping spiders (Phidippus) readily fed on the standard prey we used to maintain them in the lab if at least 30 hours had elapsed since the last feed, and this was the basis of our decision to use this period of food deprivation in the experiments. Regardless of the exact hunger levels in our experiments, we did detect significant differences in spider behaviour towards different stimulus types, demonstrating their ability to perceive and respond to different levels of mimicry.

For the crab spiders, the period of deprivation was a minimum of 48 hours, but in practice it is likely to have been considerably longer: these spiders were collected from the wild, sitting in wait for prey on flowers (implying they were already motivated to hunt); individuals found with recently killed prey were not included. Observations in the field (Ajuria and Reader, unpublished) suggest Synema successfully kills prey once every few days, so the median time since the last meal in the experiment was likely around 3-5 days. We have now clarified this in the methods section (lines 991-997). For comments on crab spider behaviour in the trials, see our response below.

3. In the Discussion, I miss information on other signals that mimics and their models are producing (buzzing, smell) which are more important for many arthropod predators than vision. These signals might be more aposematic and informative of prey type than visual signals and alert the predators. Put in another way, experienced predators, such as spiders used in this study, may perceive dangerous wasp only if the visual stimulus is accompanied by sound signals.

We have added some discussion of non-visual signals (line 493-496) and altered our conclusion of the Invertebrate Predators section to focus on visual signals (line 476), given that, as the referee points out, other signals may modify the results in the case of real prey.

Specific comments:

4. Lines 64-65: This is only one solution but limited because predators and their perception have also evolved.

We agree that selective pressures acting on species that existed in the past may have been different. Our comment was not intended to evoke the idea of reconstructing historical phenotypes (although that would certainly be a potential use of our techniques), but rather is intended to apply more generally to any hypothetical phenotypes that fill in currently unoccupied parts of the adaptive landscape. We have adjusted our wording to clarify this (line 66-68).

5. Lines 66-68: The statement needs to be supported by a citation.

We have added supporting references (line 72).

6. Figures 2, 3: The ordinate is on a logit scale which is difficult to read as it has no bounds. Transform it to probability (using antilogit).

We used the logit transformation to reduce heteroskedasticity which would impact our analysis. We can understand the referee's rationale for preferring plots to show the untransformed data, since the scale then runs from 0 to 1. This comes with a note of caution though, which is that our "level of protection" metric is relative rather than absolute, as rankings depend on the selection of stimuli available to the predators within a given session. Therefore 0 does not mean "no protection" and 1 does not mean "full protection", and meaningful comparison relies on the relative positions of different phenotypes rather than their absolute values, the former being equally interpretable whether or not the transformation is used.

A further limitation arises when plotting the learning curves (Extended Data Figures 1 and 2), which is that the curves must be fitted to the transformed data (for reasons stated above). It is possible to back-transform the curves to fit the untransformed data, but this causes a poor visual fit between the curves and the summary data (mean \pm confidence interval).

For those reasons we would prefer to keep the plots scaled as they are, but if the editor and/or referees feel strongly that the untransformed data would be clearer, we are happy to make the change, though we might need also to present the transformed data in order to show the learning curves effectively.

7. Figure 3. In both experiments wild great tits were used yet the responses were different for the same stimuli. For example, M100 had values below -1 (on logit scale) in the first experiment but about -0.5 in the second experiment. What has happened, why such a big discrepancy? Explain it.

The reason behind this discrepancy becomes clearer when looking at the learning curves, which we include in the revised submission. See our response to R2

point 1 for more details. We have now added discussion of the reasons behind this difference (line 286-297).

8. Lines 313-314: Why non-parametric test was used? Clearly, the high number of presentations shows there were repeated use of chicks which should be treated as random effects in time to event analysis.

Indeed, the referee is correct that by using a simple non-parametric test for this quick comparison we did not take into account the repeated measures within individuals (in contrast to our main analysis, which did use mixed models). We have revised this to use a linear mixed model, treating chick as a random effect (line 369). The outcome of the analysis is unchanged.

9. Lines 348, 418. The Eye of the beholder hypothesis is completely inappropriate here. Replace by a paper on multiple-predator hypothesis.

With respect, we disagree that the Eye of the Beholder is irrelevant, for the reasons discussed under R3 point 1 above, so we have kept this reference. However, we agree that the Multiple Predator hypothesis is also relevant here, and have now highlighted this (line 411-412, 512-515).

Lines 400-404: The authors seem to completely ignore other senses of arthropod predators to explain the observed pattern, yet these are often more important than vision. Namely the olfaction and seismic senses are very important. In addition, the movement even though perceived by vision plays a significant role.

We agree that in real encounters, predators may have non-visual information available to them, and that this could be important to arthropod predators. We have added a mention of this point in the main discussion (line 493-497) – see also our response to R3 point 3. Here, olfaction or sound cannot explain the observed results since those traits did not vary among our stimuli, and we know that the predators are responding to differences in visual signals. To clarify, we have changed our wording to emphasise that our conclusions focus on visual features only (line 476).

Line 754: Only here the reader is informed that the data presented earlier come from both blue and great tit. But on line 216 only great tits are mentioned.

We have corrected this inaccuracy in the main text (line 248-249).

Line 367: I was particularly curious to see how the authors could make a sit-and-wait predator (Synema spiders) to attack a plastic mimic. From the video I could see that the spider is reaching out for a 3D model but I doubt this is a grabbing behaviour resembling an attack. It rather looked like a behaviour to escape.

Behavioural assessment was based on a previously established ethogram (Morris and Reader 2016) and the distinction between aggressive and escape behaviours was usually clear to observe. We did observe clear instances of attack behaviour, which seemed to be triggered by vertical movement of the stimulus implemented by our Arduino motor. Unfortunately, we do not have videos from most trials and the footage included in the supplementary information came from a preliminary trial with a spider which had not been deprived of food for 48 hours, in which we agree the spider's behaviour is not particularly clear cut (and would not have been classed as an attack had it occurred in the main trials). This is important contextual information we should not have overlooked – we have now added some explanatory text to the video description (lines 1172-1181).

Crucially, regardless of the exact motivation behind the observed behaviour, we did detect significant differences according to stimulus type, which can only mean that the spiders are able to discriminate visually among at least some of the stimuli, and are affected by the level of mimetic accuracy.

Literature cited:

- Davies, N. B. (1977). "Prey selection and the search strategy of the spotted flycatcher (*Muscicapa striata*): A field study on optimal foraging." *Animal Behaviour* 25: 1016-1033.
- Erichsen, J. T., J. R. Krebs and A. I. Houston (1980). "Optimal Foraging and Cryptic Prey." *Journal of Animal Ecology* 49(1): 271-276.
- Lindström, L., R. V. Alatalo, A. Lyytinen and J. Mappes (2004). "The effect of alternative prey on the dynamics of imperfect Batesian and Müllerian mimics." *Evolution* 58(6): 1294-1302.
- Morris, R. L. and T. Reader (2016). "Do crab spiders perceive Batesian mimicry in hoverflies?" *Behavioral Ecology* 27(3): 920-931.
- Pyke, G. H., H. R. Pulliam and E. Charnov (1977). "Optimal foraging: A selective review of theory and tests." *Quarterly Review of Biology* 52: 137-154.
- Sherratt, T. N. (2002). "The evolution of imperfect mimicry." *Behavioral Ecology* 13(6): 821-826.
- Warton, D. I. and F. K. C. Hui (2011). "The arcsine is asinine: the analysis of proportions in ecology." *Ecology* 92(1): 3-10.

Appendix: extra analysis of Discrimination Ability experiment

Referee 2 asks (point 3 above): “What do the results look like if one removes the non-mimetic flies and wasps entirely from the sequence which drive the first and last choices respectively: is there still strong evidence of birds prioritizing M75V25 over M25V75 say?”

We carried out a supplementary analysis of the testing phase data (filtered to exclude the first 9 days when learning was ongoing, as before). We filtered out all dishes with M100, C100, S100 or V100 stimuli from the feeding sequence for each feeder and session. We then converted this new, shortened sequence to a set of protection values from 0 to 1, corresponding to the first and last dishes of the new sequence respectively, and logit transformed the resulting values. We repeated the same analysis as conducted for the main dataset, fitting a linear model (preference ~ (reward + reward:(transect*phenotype) + edge)*feeder) and comparing to nested sub-models. The model with all terms was the best supported with $\Delta AIC = 7.11$ to the next best model. As before, Tukey’s post-hoc comparisons were used to test for differences among different levels of the phenotype and transect variables. The table below shows selected post-hoc contrasts (contrasts comparing two different transects, such as M 75% fly – S 50-50, are not displayed for the sake of simplicity). Note that all displayed contrasts are strongly significant except for M 75% fly - M 50-50 and S 75% fly - S 50-50. These results qualitatively match those from the main analysis (see groupings in Figure 2B).

contrast	estimate	SE	df	t.ratio	p.value
M 75% fly - M 50-50	-0.10	0.213	1097	-0.47	1.000
M 75% fly - M 75% wasp	-1.80	0.213	1097	-8.43	<0.001
M 50-50 - M 75% wasp	-1.70	0.213	1097	-7.96	<0.001
S 75% fly - S 50-50	-0.40	0.214	1097	-1.86	0.639
S 75% fly - S 75% wasp	-1.90	0.214	1097	-8.87	<0.001
S 50-50 - S 75% wasp	-1.50	0.215	1097	-6.95	<0.001
C 75% fly - C 50-50	-0.95	0.213	1097	-4.46	<0.001
C 75% fly - C 75% wasp	-1.77	0.214	1097	-8.28	<0.001
C 50-50 - C 75% wasp	-0.82	0.214	1097	-3.84	0.004

Dear Dr. Gee,

We thank the referees for their positive comments on our latest draft of the manuscript, and we are pleased that our revisions addressed their original comments. The new comments have been very helpful in improving the manuscript further, especially with respect to clarifying some points related to the statistical analysis. Below we respond to each of the points in turn. Line numbers refer to the “all markup” version of the manuscript that includes tracked changes from the previous draft.

Yours sincerely,

Christopher Taylor, on behalf of the authors

Referee #2 (Remarks to the Author):

i) From the extended data table 2 it appears (if the base formula has edge as the only predictor) that the authors compared the AIC of models that included interactions (such as $m:p$) but lacked the composite main effects (notably p). This violates the principle of marginality and in so doing generates incorrect estimates of the interaction coefficients (since the interaction term now includes the main effects it is composed of). The lack of hierarchical considerations when fitting models may or may not matter when it comes to model comparison (it may simply be a re-parameterized model), but it is sufficiently important to warrant checking (see Grafen & Hails 2002 Modern Statistics, p. 192). Indeed, some software prevents analysts from fitting models with interactions without all their constituent main effects.

We thank the referee for their attention to detail here! This concern would be valid if the “phenotype” (p) and “mealworm” (m) effects were fully crossed, giving 10 possible combinations. In our experiment, the wasp phenotype always includes a mealworm and the others never do, and so there are only 5 possible combinations, which are fully specified by the $m + m:p$ effects. Inclusion of a separate phenotype main effect would be redundant in this situation. We used this formulation in order to facilitate comparison between $(m + m:p)$ and (m) , the latter model being nested within the former, testing whether phenotype is only relevant when comparing wasp and not-wasp (i.e. the mealworm effect), or is important across the range of mimetic phenotypes.

*On reflection, and in light of the referee’s comment, our original approach was unnecessarily complicated. A simpler version of the model in which mealworm is omitted and phenotype is treated as a simple 5-level factor (level of protection ~ (phenotype*axis + edge)*feeder) is almost exactly equivalent – it yields the same AIC values as the original version, but with the omission of some rows, specifically those that isolate the effect of wasp vs not-wasp from the effect of accuracy within the*

mimics, or from the effect of axis. However, these aspects are covered by our post-hoc tests, so no information is lost by using this simpler version.

We have edited our Materials and Methods to reflect this adjusted modelling approach (lines 1096-7) and provide the updated results of model comparison in ED Table 2. In Appendix 1 to this document, we demonstrate that the results of the new model formulation are identical to a subset of those from the original formulation that we presented in previous versions of this manuscript.

ii) On the subject of interactions, why 2 df with the intermediate:distance and yet 1 df with the treatment:distance?

Treatment is included in the base model, so adding treatment:distance adds 1 df, whereas intermediate is not in the base model, so adding in the interaction also requires adding the main effect. This was not made clear by our description in ED Table 3 which lumped together main effects and interactions. We have now edited ED Table 3 to show main effects and interaction terms separately.

iii) Given the potential role of “edge” (outer perimeter), we assume that treatments were randomised with respect to edge?

Yes, treatments (i.e. stimuli) were randomised across all board positions, which meant that each treatment had equal probability of appearing in the edge. We have tweaked wording (line 887) to clarify this.

iv) A rose by any other name smells just as sweet, but on second reading we were also temporarily confused by the term “transect”, which is suggestive of stimuli being laid out in a line.

We have removed all uses of the term “transect” to avoid confusion (e.g. line 78), instead using “axis” where appropriate (e.g. line 106)

Referee #3 (Remarks to the Author):

The multiple models hypothesis seems to be misinterpreted here. You predict greater protection for an intermediate mimic (l. 231). This is not in agreement with Edmunds (1991) who states that the poor (=intermediate) mimic gains higher protection NOT on the individual BUT population level. In other words, an individual of a poor mimic population will have lower (NOT HIGHER as you predict) protection from predators compared to a good mimetic individual because it resembles more models so the

population of the poor mimic can occur on a larger spatial and/or temporal scale. So the whole population (NOT an individual) of a poor mimic will have higher fitness than the population of a good mimic. To test this hypothesis per se you would need to compare POPULATIONS of poor and good mimics in temporal and spatial scale(s). This was not done in your experiment as you only compared resemblance to two models. Yet, even your experiment has some merit for the Multiple models hypothesis. The results of your experiment, in contrast to what you conclude, support the Multiple models hypothesis, in my opinion, because the protection of intermediate (poor) mimics was not significantly different from protection of good mimic. So you would need to reformulate the predictions and conclusions of this experiment.

We agree with the reviewer that our predictions around the Multiple Models hypothesis could have been clearer. We do already state that we predict an increase in fitness for an intermediate relative to the fitness that would be received by mimic of similar accuracy towards a single model (lines 226, 301, 477). That is to say, we predict a flatter region of the adaptive landscape between the two models than “outside” them, as predicted under some circumstances by Sherratt (2002; see equation 8). We do not necessarily predict a hump of higher fitness for an intermediate mimic above the levels of a perfect mimic, which we agree is not predicted in existing theoretical models (though it is suggested by some authors, e.g. McLean et al 2019 <https://doi.org/10.1086/706769>). For clarity, we now add explicit acknowledgement that theory does not support an advantage for intermediate over perfect mimics (line 225); we have removed two sentences stating that we have not found an empirical advantage for intermediate over perfect mimics (lines 303 and 509) which, while true, might mislead a reader into thinking this was the main hypothesis being tested; and we have added reference to Edmunds (2000) in addition to Sherratt (2002) where we discuss the effect of integrating across spatial and temporal scales (line 308).

L. 527: Or they use different stimuli – acoustic, seismic or chemical.

True, we have now clarified that our conclusions apply only to mimicry of visual features (lines 511 and 516); our experiments do not address the possibility of mimicry in other sensory modalities.

I looked at the statistical analyses in more detail and I have the following comments. Overall, the analyses appear to be correct. I use the word appear because not all explanatory variables are defined (namely, reward, edge, stimulus, treatment) thus it was not possible to check whether the linear predictor of the fitted models make sense. I believe they do, though.

We originally defined each variable within the “Statistical Analysis” section the first time it is used, but as this information was dotted through the section it may have been hard to get an overview. We have now rearranged the text so that all variables are defined in a single block for each experiment prior to detailing the model formulae (lines 1067-1070, 1089-1096 etc.); this arrangement should provide greater clarity.

Further, I was surprised why authors estimated confidence intervals using bootstrapping. If the design of the experiments was completely random without (dependencies) pseudoreplications then the use of bootstrap would be fine. But the design was more complicated (nested), then the bootstrap estimates might not be better than estimates from the linear models. It depends how bootstrapping was done. Did authors did not believe the estimates (mean values and their standard errors) of the model? They do not describe nor explain why bootstrapping was used.

*We originally chose to base our confidence intervals on bootstrapping because this approach is broadly applicable and minimises assumptions about the distributions of the data. However, the referee is correct to point out that it would be more appropriate and consistent to calculate confidence intervals based on the same distributions used in our modelling. We have therefore adjusted our plots to show confidence intervals based on the t -distribution (approximating to $1.96*SE$ where sample sizes are large), with appropriate transformations in the cases where non-Gaussian GLMs were used. To show the nested structure of the wild bird experiments, where each feeder had multiple copies of each stimulus type, we now plot mean and confidence intervals for each feeder separately, along with the overall mean for each phenotype (as we did originally) as a summary to highlight trends (Figures 2 and 3). Our other experiments have less complicated structure with each individual only receiving a given stimulus type once, so we now show each individual data point along with mean (as before) and confidence intervals (now based on SE) as a summary (Figures 4 and 5).*

The protection values were logit transformed which is OK. However, then it is not clear which family has been used in GLM analysis. Such values are not from binomial distribution so the use of binomial family is not suitable and beta should be used. Beta is, however, not implemented in many packages and from the text of methods it is not clear which package for GLM and GLMM authors used (l. 1057).

The analyses in question (Discrimination Ability and Multiple Models experiments) used Gaussian family GLMs (i.e. simple LMs) after transformation of the data. We found this to be effective in minimising heteroskedasticity of the data and ensuring normality of residuals. We have clarified the family of model used in each case (lines 1096, 1133). Models were fitted using the package lme4, to which we have now added a reference (line 1050) – thank you for pointing out that omission.

The authors fitted two models using NLS regression which is not trivial to use. Indeed, for some data the algorithm did not converge. However, I miss arguments why these two models were used. Typically, this is done because some (or all) model parameters have biological meaning. But this was not the case which makes me wonder why they did not use linear models for curvilinear relationships, instead, which are easier to fit. The sigmoid 3-parameter model (l. 1082) could be replaced by 2-parameter sigmoid (logistic) model implemented within GLM and binomial errors. Similarly, the 3-parameter asymptotic model could be replaced by 2-parameter (Michaelis-Menten) model fit with GLM with Gamma errors.

The main purpose of our NLS regression, aside from illustration of trends, was to identify a period of learning after which bird preferences were relatively stable (see lines 156-9). This allowed us to focus our analysis on the behaviour of educated, rather than naïve, birds. The learning period is defined by three key characteristics, each with a biological interpretation: a starting level of discrimination, a rate of change and a final level of discrimination. None of these characteristics could be accurately predicted in advance, and indeed their values varied according to stimulus type (see ED figure 1b). While we agree that alternative formulations would be possible, our use of NLS allowed us to derive the learning curve characteristics empirically, whereas fitting a model with fewer parameters (such as the GLM methods suggested) inevitably requires extra assumptions. While our NLS models were less effective in fitting data from the Multiple Models testing phase, the limited clarity of temporal trends within the data (ED figure 2b) mean that the learning period would be difficult to pin down empirically regardless of model choice. Therefore, we felt the most appropriate choice here was to maintain consistency with the Discrimination Ability analysis and again choose day 10 as a cut-off. We have inserted some justification of our model choice into the manuscript (lines 1082-5)

L. 1119: How did you deal with feeders in the GLM models? On l. 1099 you state that feeder was fitted as fixed effect due to low number of feeders (I agree) but then on l. 1119 you state it was fitted as random effect, yet the number of feeders was the same.

In the Multiple Models experiment we had six feeders rather than three (see line 812), which we judged enough to fit as a random effect (line 1125).

Appendix 1. Comparison of models fitted to data from the testing phase of the Discrimination Ability experiment. Here, we show results of the model `level of protection ~ (mealworm + mealworm:(axis*phenotype) + edge)*feeder` which was the model fitted in the original manuscript and shown in the original version of ED Table 2 (renaming “transect” to “axis”). White rows are direct equivalents of rows in the new version of the analysis now shown in ED Table 2, which is `level of protection ~ (phenotype*axis + edge)*feeder`. These rows have the same df and AICc values as shown in ED Table 2 and just with the “additional terms” specified in a different way. Grey shaded rows are not included in the new version of the analysis.

Model rank	df	Log Likelihood	AICc	ΔAICc	additional terms*
1	43	-4602.2	9291.6	0.0	f + m + a:m + e:f + f:m + m:p + a:f:m + a:m:p + f:m:p + a:f:m:p
2	25	-4625.4	9301.2	9.6	f + m + a:m + e:f + f:m + m:p + a:f:m + a:m:p
3	31	-4621.8	9306.3	14.7	f + m + a:m + e:f + f:m + m:p + a:f:m + a:m:p + f:m:p
4	21	-4637.0	9316.3	24.7	f + m + a:m + e:f + f:m + m:p + a:m:p
5	41	-4618.9	9321.0	29.4	f + m + a:m + f:m + m:p + a:f:m + a:m:p + f:m:p + a:f:m:p
6	27	-4633.5	9321.6	30.0	f + m + a:m + e:f + f:m + m:p + a:m:p + f:m:p
7	19	-4643.2	9324.7	33.1	f + m + a:m + e:f + f:m + m:p + a:f:m
8	25	-4639.7	9329.8	38.2	f + m + a:m + e:f + f:m + m:p + a:f:m + f:m:p
9	23	-4642.7	9331.9	40.3	f + m + a:m + f:m + m:p + a:f:m + a:m:p
10	29	-4638.8	9336.2	44.6	f + m + a:m + f:m + m:p + a:f:m + a:m:p + f:m:p
11	15	-4654.7	9339.6	48.0	f + m + a:m + e:f + f:m + m:p
12	13	-4656.8	9339.8	48.2	f + m + e:f + f:m + m:p
13	19	-4652.7	9343.6	52.0	f + m + a:m + e:f + m:p + a:m:p
14	21	-4651.3	9344.9	53.3	f + m + a:m + e:f + f:m + m:p + f:m:p
15	19	-4653.4	9345.1	53.5	f + m + e:f + f:m + m:p + f:m:p
16	19	-4654.0	9346.2	54.6	f + m + a:m + f:m + m:p + a:m:p
17	25	-4650.1	9350.7	59.1	f + m + a:m + f:m + m:p + a:m:p + f:m:p
18	17	-4661.0	9356.3	64.7	f + m + a:m + f:m + m:p + a:f:m
19	23	-4657.1	9360.7	69.1	f + m + a:m + f:m + m:p + a:f:m + f:m:p
20	13	-4670.2	9366.5	74.9	f + m + a:m + e:f + m:p
21	11	-4672.3	9366.7	75.1	f + m + e:f + m:p
22	11	-4674.0	9370.1	78.5	f + m + f:m + m:p
23	13	-4672.2	9370.4	78.8	f + m + a:m + f:m + m:p

24	15	-4670.3	9370.8	79.2	m + a:m + m:p + a:m:p
25	17	-4670.0	9374.2	82.6	f + m + a:m + m:p + a:m:p
26	17	-4670.2	9374.6	83.0	f + m + f:m + m:p + f:m:p
27	19	-4668.4	9375.0	83.4	f + m + a:m + f:m + m:p + f:m:p
28	7	-4690.1	9394.3	102.7	m + m:p
29	9	-4688.3	9394.6	103.1	m + a:m + m:p
30	9	-4689.8	9397.7	106.1	f + m + m:p
31	11	-4688.0	9398.1	106.5	f + m + a:m + m:p
32	16	-4779.5	9591.1	299.5	f + m + a:m + e:f + f:m + a:f:m
33	12	-4790.1	9604.4	312.8	f + m + a:m + e:f + f:m
34	10	-4797.5	9615.0	323.4	f + m + e:f + f:m
35	14	-4795.8	9619.8	328.2	f + m + a:m + f:m + a:f:m
36	10	-4804.3	9628.6	337.0	f + m + a:m + e:f
37	10	-4806.2	9632.4	340.8	f + m + a:m + f:m
38	8	-4811.5	9639.1	347.5	f + m + e:f
39	8	-4813.2	9642.4	350.8	f + m + f:m
40	6	-4821.0	9654.0	362.4	m + a:m
41	8	-4820.7	9657.4	365.8	f + m + a:m
42	4	-4827.9	9663.8	372.2	m
43	6	-4827.6	9667.2	375.7	f + m
44	7	-5792.6	11599.2	2307.6	f + e:f
45	3	-5806.4	11618.8	2327.2	
46	5	-5806.3	11622.5	2331.0	f

* All models are based on the initial formula (level of protection ~ edge), with the addition of the listed terms. Abbreviated terms: e = edge, f = feeder, p = phenotype, and a = axis (replacing t = transect from previous drafts).